# Towards a molecular picture of the archaeal cell surface

Matthew C. Gaines[1,2,10], Michail N. Isupov [3,10], Mathew McLaren [1,2,10], Clara L. Mollat[4,10], Risat Ul Haque[1,2], Jake K. Stephenson [1,5], Shamphavi Sivabalasarma[4,6], Cyril Hanus [7], Daniel Kattnig [1,8], Vicki A. M. Gold [1,2], Sonja Albers [4,6,9] & Bertram Daum[1,2] ✉

Archaea produce various protein filaments with specialised functions. While some archaea produce only one type of filament, the archaeal model species *Sulfolobus acidocaldarius* generates four. These include rotary swimming propellers analogous to bacterial flagella (archaella), pili for twitching motility (Aap), adhesive fibres (threads), and filaments facilitating homologous recombination upon UV stress (UV pili). Here, we use cryo-electron microscopy to describe the structure of the *S. acidocaldarius* archaellum at 2.0 Å resolution, and update the structures of the thread and the Aap pilus at 2.7 Å and 2.6 Å resolution, respectively. We define features unique to archaella of the order Sulfolobales and compare their structure to those of Aap and threads in the context of the S-layer. We define distinct N-glycan patterns in the three filaments and identify a putative O-glycosylation site in the thread. Finally, we ascertain whether N-glycan truncation leads to structural changes in archaella and Aap.

Archaea generate a variety of cell-surface filaments that are crucial to their biology. Many of those filaments share a common origin with bacterial Type-IV pili (T4P)[1–4]. These include archaeal flagella, known as archaella[5–8], used for swimming motility, and pili functioning in adhesion, twitching motility[9–12] and homologous recombination[13,14]. Archaea also express filaments distinct from T4P, such as the threads of *S. acidocaldarius*[15] and the hami of *Altiarchaeon hamiconnexum*[16]. The latter two have been implicated in adhesion and biofilm formation, however their structures are distinct from each other. Whereas threads are structurally similar to bacterial type-I pili (T1P)[17,18], hami have a unique barbed-wire-like architecture[16].

Several species of Archaea produce more than one filament. The archaeal model organism *Sulfolobus acidocaldarius* is a prime example. *S. acidocaldarius* expresses four filaments: archaella[5–8], adhesive A-pili (Aap)[9,10], threads[15] and UV-pili (Ups)[13,14].

Archaella and Aap are helical filaments consisting of archaellins or pilins, respectively. Due to their common origin with bacterial T4P, their subunit proteins obey the typical T4P blueprint, characterised by an N-terminal α-helix, followed by a β-strand rich globular C-terminal domain. While the α-helical tails bundle to assemble the core of the filaments, the globular c-termini face outside and thus form an outer sheath[7,19].

¹Living Systems Institute, University of Exeter, Exeter, UK. ²Department of Biosciences, Faculty of Health and Life Sciences, University of Exeter, Exeter, UK. ³Henry Wellcome Building for Biocatalysis, Department of Biosciences, Faculty of Health and Life Sciences, University of Exeter, Exeter, UK. ⁴Institute of Biology, Molecular Biology of Archaea, University of Freiburg, Freiburg, Germany. ⁵School of Natural Sciences, Faculty of Faculty of Environment, Science and Economy, University of Exeter, Exeter, UK. ⁶Spemann Graduate School of Biology and Medicine, University of Freiburg, Freiburg, Germany. ⁷Institute of Psychiatry and Neurosciences of Paris, Inserm UMR1266 -Université Paris Cité, Paris, France. ⁸Department of Physics and Astronomy, Faculty of Environment, Science and Economy, University of Exeter, Exeter, UK. ⁹Signalling Research Centres BIOSS and CIBBS, Faculty of Biology, University of Freiburg, Freiburg, Germany. ¹⁰These authors contributed equally: Matthew C. Gaines, Michail N. Isupov, Mathew McLaren, Clara L. Mollat. ✉ e-mail: b.daum2@exeter.ac.uk

The archaellum is a rotary filament comprised of thousands of helically organised archaellins[5,7]. Clockwise rotation of the archaellum propels the archaeal cell forward, and anticlockwise rotation backward[5]. Like T4 pilins, archaellins are initially expressed as membrane proteins through the Sec translocon and subsequently cleaved by a type-III signal peptidase (pibD in archaea). This processing step facilitates their assembly into the archaellum filament[20,21] (Supplementary Fig. 1).

The assembly and gyration of the archaellum is powered by a multiprotein motor complex that is embedded in the cellular membrane[7] (Supplementary Fig. 1). This complex is typically encoded in the same cluster as the archaellin subunits[6], and similar gene clusters have been located across countless archaeal species, including the arguably most studied archaeal phyla; the euryarchaeota and the crenarchaeota.

At the heart of the motor complex sits a central protein, ArlJ[22]. ArlJ is a homologue of the bacterial T4P platform PilC[23,24], and thus thought to drive archaellum assembly and rotation. ArlJ is powered by ATP hydrolysis via the action of a single cytoplasmic ATPase (ArlI). A KaiC-like protein (ArlH) interacts with ArlI in an autophosphorylation-dependent manner, possibly acting as a regulator that switches the motor from archaellum assembly to rotation[22]. Surrounding the central ArlJ/ArlI/ArlH complex is a ring composed of ArlX in crenarchaeota and ArlC/D/E in euryarchaeota. ArlX forms rings in various diameters in vitro, and probably surrounds the archaellum motor complex in vivo. Euryarchaea have ArlCDE instead of ArlX, which is proposed to couple the archaellum machinery to chemosensory pathways[8,25–27]. ArlF and ArlG form the stator complex. ArlG assembles into an open helical filament mimicking the interaction interface of ArlB[28]. This filament is capped by ArlF which anchors the stator complex in the cell envelope by interaction with an S-layer[22,28] (Supplementary Fig. 1).

Depending on the species, 1–5 archaellins are encoded in the archaellum cluster[29]. While in many species the archaellum filament is composed of only one archaellin[30–32], others such as that of the euryarchaeon *Methanocaldococcus villosus*, consist of two alternating archaellins[33]. Any super-numerous subunits that are encoded in the genome but are not a part of the main filament are referred to as minor archaellins and may either form nucleating or capping structures of archaella[33–37], or act as hooklike structures[38,39]. In *Sulfolobus acidocaldarius*, only one (major) archaellin is known[40].

In addition to the archaellum, *S. acidocaldarius* generates two non-rotary T4P-like filaments - the Aap and Ups. While the Ups is only expressed upon UV stress and aids recombination for genome repair[14], the Aap is the most common filament of this species under non-stressed conditions[10]. A recent study uncovered that Aap can function in twitching motility−a cellular surface-bound motion enabled by cycles of rapid extension and retraction of pili[12]. By solving the structure of the Aap, we recently determined that the major subunit of the Aap is AapB[9,41]. Aap share the same PibD-dependent assembly pathway as archaella[10,42] and are assembled by a homologous platform protein (AapF)[9,10]. Assembly and retraction of these Aap is powered by a single ATPase AapE[9,10], which is homologous to ArlI[26]. The operon also contains an additional protein, AapX, the function of which is currently unknown.

The fourth filament, the thread, is abundant on the *S. acidocaldarius* cell surface[15]. The thread is assembled in a PibD-independent manner and thus does not belong to the superfamily of T4P-like filaments[15]. Through cryoEM and visual proteomics, we previously determined that the thread consists of multiple copies of the subunit Saci_0406, and resembles, but is not homologous to, bacterial Type-I (chaperone-usher; CU) pili[17,43]. Saci_0406 consists of a β-strand rich globular c-terminus and an extended N-terminal β-strand. Like in CU pili, the N-terminal strand inserts into a downstream subunit, thereby completing a β-sheet via donor strand complementation (DSC). The subunits additionally form intermolecular isopeptide

bonds, thus covalently interlinking the thread into a strong, chain-like fibre[15]. The gene cluster encoding for Saci_0406 also contains genes for Saci_0405, a putative filament capping protein. Further, the cluster contains the genes *saci_0404*, *saci_0407* and *saci_0408*. While the function of these genes is elusive, it is assumed that they are part of the thread assembly machinery[15].

Archaeal cell surface proteins, including cell-external filaments and S-layers, are usually highly glycosylated. N-linked and O-linked glycosylation exist in archaea and while the former is strictly associated with NXS/T motifs, the consensus sequences for the latter are not known[44,45]. Glycosylation generally aids protein folding, as well as their assembly into functional complexes[45]. Archaeal surface glycans have been proposed to be crucial for the stability of S-layers[46], and filaments[7,47], and to provide resilience towards extreme environmental temperatures and pH values[45,48,49]. In addition, surface glycans also play a role in host-virus interaction[50], and provide "recognition tags" that ensure species-selective mating[13]. The structure, composition and sequence of those glycans are highly variable across the phylogenetic tree, and each archaeal species usually produces its own unique N-glycan[44,45].

*S. acidocaldarius* produces N- and O- linked glycans. The complex composition of the N-linked glycan has been determined by mass spectrometry[51], and structurally characterised by cryoEM[15]. The N-glycan is a tribranched hexasaccharide, containing two N-acetyl glucosamine (GlcNac), two Mannose (Man), one Glucose (Glc) and one 6-sulfoquinovose (QuiS) molecule. The O-linked glycan, on the other hand, appears to be a simple mannose modification[52].

It has been shown that partial or full ablation of glycans in *S. acidocaldarius* drastically impacts swimming motility[40]. A deletion mutant MW039 (*Δagl3*), lacking the sulfoquinovose synthase Agl3 generates a trisaccharide N-glycan, missing the sulfoquinovose, one mannose and one glucose molecule[53]. Another deletion mutant MW043 (*Δagl16*) lacks the glycosyltransferase Agl16 and thus generates a pentasaccharide glycan without the terminal glucose. Strikingly, both deletion strains showed severely decreased motility− *Δagl16* retained only 4%, and *Δagl3* 0% of the wild type[40]. The observed impact of N-glycan truncation on motility poses the question whether N-glycans are required for proper filament structure.

Here we employed cryoEM to solve the structure of the *S. acidocaldarius* archaellum to complete our picture of the visible array of filamentous appendages of this archaeon under growth conditions without UV stress. With a global resolution of 2.0 Å, our map is currently the best resolved of any archaellum filament, providing insights into its structure at exquisite detail. In addition, we determined the structures of the AAP and the thread from the same organism at improved global resolution (2.6 Å and 2.7 Å, respectively), providing improved structural insights into the surface glycans and revealing a putative O-glycan modification in the thread. To tackle the long-standing question of how glycans impact on the architecture of motile filaments, we solved the structures of archaella and Aap and from the glycan truncation mutant *Δagl3* at global resolutions of 2.4 Å and 2.6 Å, respectively, and discuss our findings in the context with previous motility experiments[40]. By comparing the *S. acidocaldarius* archaellum with Aap and threads in context with our previously published atomic model of the *S. acidocaldarius* S-layer[54], we provide a hypothetical framework for S-layer-filament interaction.

## Results

### Structure of the *S. acidocaldarius* Archaellum

To solve the structure of the *S. acidocaldarius* archaellum, we cultivated a hyper-archaellated strain (MW2106, Supplementary Fig. 2) to increase the yield of our filament preparation. Archaella were sheared from the cell and then further purified through CsCl gradient centrifugation. This resulted in a sample containing archaella and a

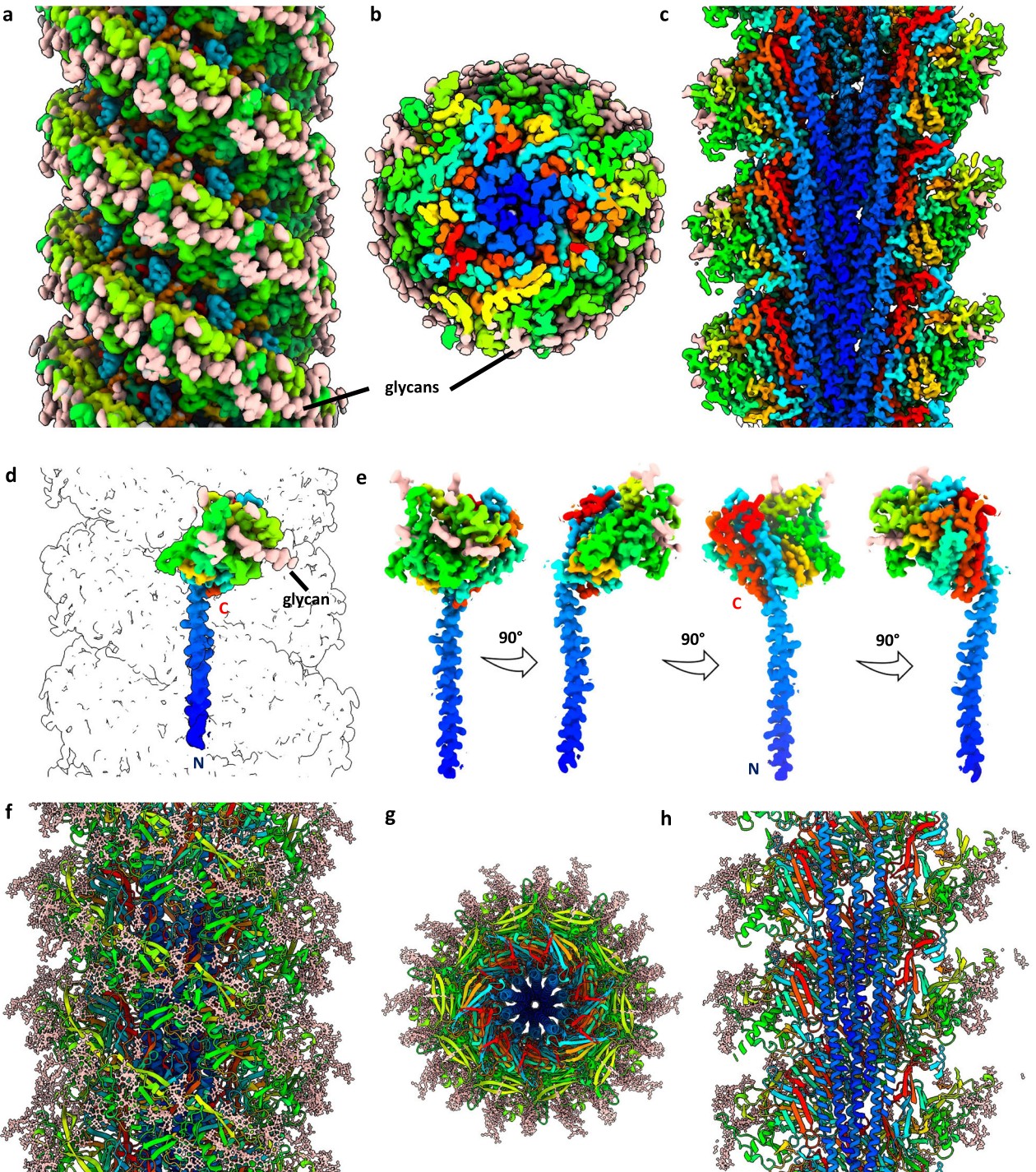

**Fig. 1 | Helical reconstruction and atomic model of the *S. acidocaldarius* archaellum.** 2.0 Å resolution cryoEM map of the archaellum in side view (**a**), end-on view (**b**) and cross-section (**c**). Rainbow colours indicate ArlB N-termini in blue and c-termini in red. Glycan densities are coloured peach. **d** cryo-EM map of a single ArlB subunit as it is positioned in the filament. **e** ArlB seen from different angles, rotated in 90° steps around the long axis of the α-helix (blue). **f**–**h** Atomic model of the archaellum with the protein backbone as rainbow ribbons and glycans as peach sticks. **f** Side view, **g** end-on view and **h** cross-section.

smaller population of threads[10] (Supplementary Fig. 3a, d). The suspension was applied to cryoEM grids and subsequently plunge-frozen in liquid ethane. Following cryoEM data collection using a Titan Krios TEM, we reconstructed a high-resolution map of the archaellum at a final average resolution of 2.0 Å using CryoSPARC[55] (Fig. 1, Supplementary Figs. 4 and 5). The map revealed a structure typical for archaella, with an α-helical core and a β-strand rich outer sheath (Fig. 1a–e).

## Structure of the archaellum

Based on our cryoEM map, we built an atomic model ab initio. The global resolution of 2.0 Å enabled us to easily recognise and model most amino acids manually, confirming that the archaellum is composed of the archaellin ArlB. During model building, we noticed that 11 residues of the protein sequence were lacking at the N-terminus, corresponding to the N-terminal leader peptide that is cleaved by PibD[7]. In accordance with this, the SignalP6 server[56] detected a high probability

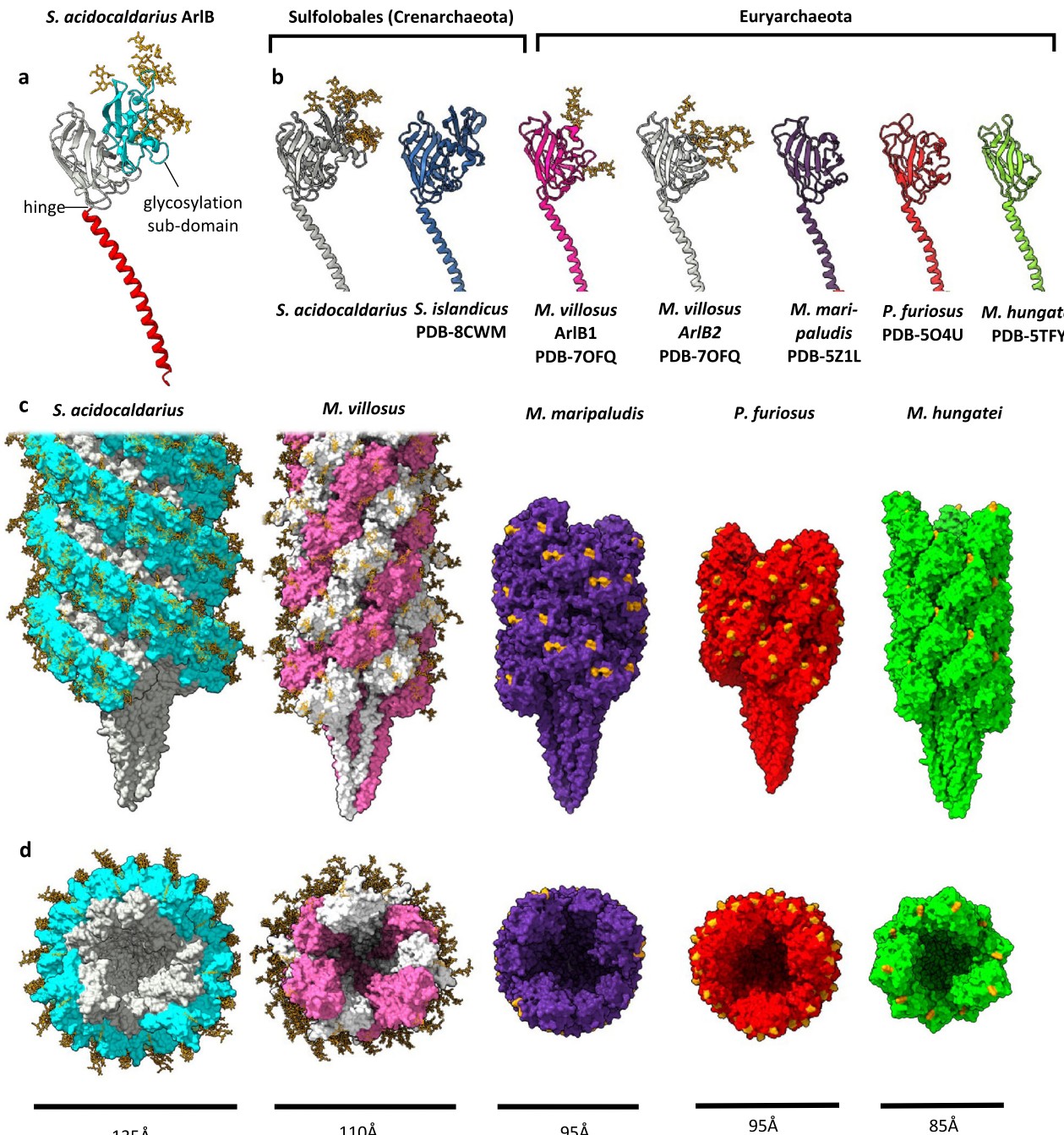

**Fig. 2 | Structures of archaella in comparison. a** structure of *S. acidocaldarius* ArlB in ribbon representation. N-terminal α-helix tail in red, globular head domain in white, and glycosylation domain in cyan. Glycans are shown as orange sticks. **b** comparison of archaellins from the crenarchaeon *S. acidocaldarius* (this study) and *S. islandicus*[57] (PDB-8CWM) with those of the euryarchaeal species *M. villosus*[33] (ArlB1 and ArlB2; PDB-7OFQ), *M. maripaludis*[32] (PDB-5Z1L), *P. furiosus*[31] (PDB-5O4U) and *M. hungatei*[30] (PDB-5TFY). While all archaella are glycosylated, they were only modelled for *S. acidocaldaius* and *M. villosus*, (orange sticks). Where glycans were not modelled, the modified Asn residues are highlighted in orange. **c, d** Comparison of assembled archaellum from *S. acidocaldarius* with those of the euryarchaeal species *M. villosus* (ArlB1 and ArlB2; PDB-7OFQ), *M. maripaludis* (PDB-5Z1L), *P. furiosus* (PDB-5O4U) and *M. hungatei* (PDB-5TFY) in side view (**c**) and end-on view (**d**). The glycosylation ridges in *S. acidocaldarius* are highlighted in cyan and glycans or glycosylated Asn residues as orange sticks. Diameters of the filaments are indicated.

of Class-III peptidase cleavage[20], which occurs between residues G11 and L12 with L12 being the first amino acid in the mature protein (Supplementary Fig. 6).

The general blueprint for T4P-like filaments applies to the *S. acidocaldarius* ArlB subunit, consisting of an N-terminal α-helix tail, connected to a globular β-strand rich (head) domain (Figs. 1d, e, 2a). The α-helix stretches between residues L15 and S58. Residues T59 and A60 form a hinge, which joins the tail to the head (Fig. 2a). The head is formed by the remainder of the protein (residues L61 to Q304). The core of the head is formed by an incomplete β-barrel consisting of 10 β-strands and obeying the typical topology for archaellins. However, after the first 5 β-strands of this barrel, (following residue L123), a pronounced sub-domain is inserted, which is comprised of a mixture of 7 β-stands and 3 α-helices. This sub-domain then joins back into 'main' head domain at T231, followed by the second set of 5 β-strands that complete the barrel (Fig. 2 a). Our *S. acidocaldarius*

archaellum structure is similar to that of *Sulfolobus islandicus*[57] (Fig. 2b). However, with our improved resolution, we were able to resolve glycan modifications in their entirety and directly answer the question of whether archaellins of Sulfolobales coordinate ions, as is the case in previously published archaella structures from euryarchaeota[32,33].

## Archaellins of Sulfolobales have a unique glycosylation subdomain

We found six dead-end densities protruding from the *S. acidocaldarius* ArlB archaellin, each originating from the asparagine residues: N145, N173, N195, N214, N218 and N229 (Fig. 2a, Supplementary Fig. 7). These densities were too large to account for side chains, showed classical structural characteristics of N-glycans and obeyed the NXS/T[15,31,33] sequon rule.

In contrast to the *M. hungatei* archaellum[30], no O-linked glycans were observed. Instead, all densities were consistent with the tri-branched hexasaccharides of the *S. acidocaldarius* N-glycan[53]. Interestingly, all the N-glycosylation sites were present within the additional sub-domain, leaving the main portion of the head devoid of glycans (Fig. 2a). Thus, we will henceforth refer to the former as glycosylation sub-domain.

For comparison, we predicted the glycosylation sites for *S. islandicus* ArlB—so far, the only other known archaellum structure from Sulfolobales and crenarchaea[57]—and found that its archaellin contains only 3 N-glycosylation sites (N176, N218, and N231). Interestingly, these three glycans coincide with homologous glycosylations sites in ArlB of *S. acidocaldarius* ArlB. On the other hand, the *S. acidocaldarius* glycosylation sites N145 and N195, are absent in *S. islandicus* (Supplementary Fig. 8). This shows that related species do not only differ in glycan composition, but also distribution pattern, aiding species recognition by the UPS system[13,14]. Nevertheless, in both species all glycosylation sites reside in the glycosylation domain.

Comparing the structure of the *S. acidocaldarius* and *S. islandicus* archaella with those of other experimentally solved for euryarchaeal species shows that, although all archaella are glycosylated, the distinct glycosylation sub-domain is unique to the *Sulfolobus* species (Fig. 2b, Fig. 3e–g, Supplementary Fig. 11). To investigate if the glycosylation sub-domain is widespread in related sulfolobales crenarchaea, we carried out extensive homology searches using SyntTax[58], and predicted the structures of ArlB homologues with AlphaFold2[59]. While always present, the glycosylation domain showed the most pronounced variability compared to the rest of the protein, which is structurally highly conserved (Supplementary Figs. 9 and 10). Moreover, the predicted glycosylation sites are also well conserved. The only outliers are *Sulfolobus acidiphilus*, and *Sulfolobus caldissimus*, where glycosylation sequons were found three residues before the C-terminus. However, as the C-termini are usually packed into the core of the filament, it is unlikely that these sequons are accessible for glycosylation (Supplementary Fig. 10).

While all euryarchaeal archaella investigated so far lack the glycosylation sub-domain, the main part of the head domain is universally present and highly conserved in eury- and crenarchaeoa (Fig. 2a, b; 3e–g, Supplementary Fig. 11, 12). This highlights that this part of the protein forms the virtually invariable archaellin core.

In the assembled *S. acidocaldarius* archaellum, the glycosylation sub-domains join up to form ridges that wind around the filament, giving it a screw-like appearance (Fig. 2c). The ridges are accentuated by the glycans, which are solely located on top of the ridges but lacking in the grooves between them (Figs. 2c, 4a). Notably, the grooves and ridges are absent in euryarchaeal archaella, which do not contain the distinct glycosylation sub-domain (Fig. 2c). In addition, the glycosylation sub-domain also gives the Sulfolobales archaella a wider diameter (Fig. 2d).

## The glycosylation domain stabilises archaellins lacking a metal binding site

Archaellins, such as those of the euryarchaea *M. villosus*[33], and *M. maripuladis*[32] contain an ion-binding loop (Fig. 3). The archaellum of *M. maripuladis* consists of multiple copies of ArlB, each of which coordinates one $Ca^{2+}$ ion[32]. Although the archaellum of *M. villosus* consists of two distinct and alternating subunits (ArlB1 and ArlB2), both contain the conserved $Ca^{2+}$-coordinating loop[33] (Fig. 3a, b). The ion-binding loop contains conserved aspartic acid and asparagine residues. Two of these aspartates co-ordinate two ionic bonds per residue, and one aspartate and the asparagine co-ordinate one ionic bond per residue (Asp154, Asp156, Asn166, Asp169 in *M. villosus* ArlB1) (Fig. 3a). Re-analysing the existing maps and model of the *P. furiosus* archaellum[31] revealed that while no metal ion was resolved in the cryoEM map, the *P. furiosus* archaellum contains the same metal coordination site (Fig. 3c). Conversely, no ion has been reported for the archaellum of *M. hungatei*[30] (Fig. 3e). The absence of this ion is likely due to the lack of the metal binding loop that is typical for euryarchaeota with larger ArlB subunits (Fig. 3f).

Careful analysis of the cryoEM map and model building of the *S. acidocaldarius* archaellum revealed no metal ion densities (Fig. 3d). Accordingly, we could not determine metal coordination sites in the deposited structure of the *S. islandicus* archaellum (PDB code: 8CWM[57]) or through multisequence alignments and Alphafold predictions of archaellin homologues from eight related species of the sulfolobaceae family (Fig. 3g; Supplementary Fig. 10).

In *S. acidocaldarius*, the consensus metal binding site is replaced by a phenylalanine, two proline, and a serine residue (F241, P242, P250, S253)—amino acid combinations that are incapable of coordinating metal ions (Fig. 3d).

Metal ion coordination is a way to provide structural integrity within a protein and indeed this has been suggested to be the case for archaella[32,33]. In the Sulfolobales lacking the metal binding site, structural integrity may be provided by an additional β-hairpin (Phe150-Tre161), which is a part of the glycosylation domain and packs against the loop lacking the stabilising ion (Fig. 3g–i).

## Structural traits of archaella, Aap and threads

Aap and archaella belong to the superfamily of Type-IV like filaments. While the Aap have retained the ability to facilitate surface adhesion and twitching—traits that are shared with some bacterial T4P[1,19,60] - archaea have evolved the archaellum, a rotary swimming T4P-like filament[5–8]. It is thought that this evolutionary divergence within T4P and the emergence of the archaellum is mainly due to the development of a distinct motor complex capable of driving rotary propulsion[19], as well as the incorporation of stator components and scaffolding proteins[8]. How these developments impact filament structure is not fully clarified.

Solving the structures of two different members of the Type-IV like filament superfamily (archaellum and Aap), as well as one distinct Type-I-like filament (thread) from the same organism, presents the perfect opportunity to study these adaptations without having to account for species-dependent differences. To facilitate this analyis, we improved the resolution of the structures of the thread and Aap from the previously published resolutions of 3.5 Å[15] and 3.2 Å[9], to 2.7 Å and 2.6 Å, respectively (Fig. 4d, g; Supplementary Figs. 4 and 5).

Although Aap and archaella both belong to the superfamily of Type IV-like filaments, they have distinct structural characteristics that aid their function. The most apparent differences are their diameters. Measuring 110 Å, the archaellum is considerably wider than the Aap (80 Å), which is mostly due to the larger head domain of the archaellin (Fig. 4a, b, d, e). The greater diameter of the archaellum may be an adaptation to withstand higher sheering and torsional forces that occur during the gyratory motion, forces that do not occur in the twitching Aap[9].

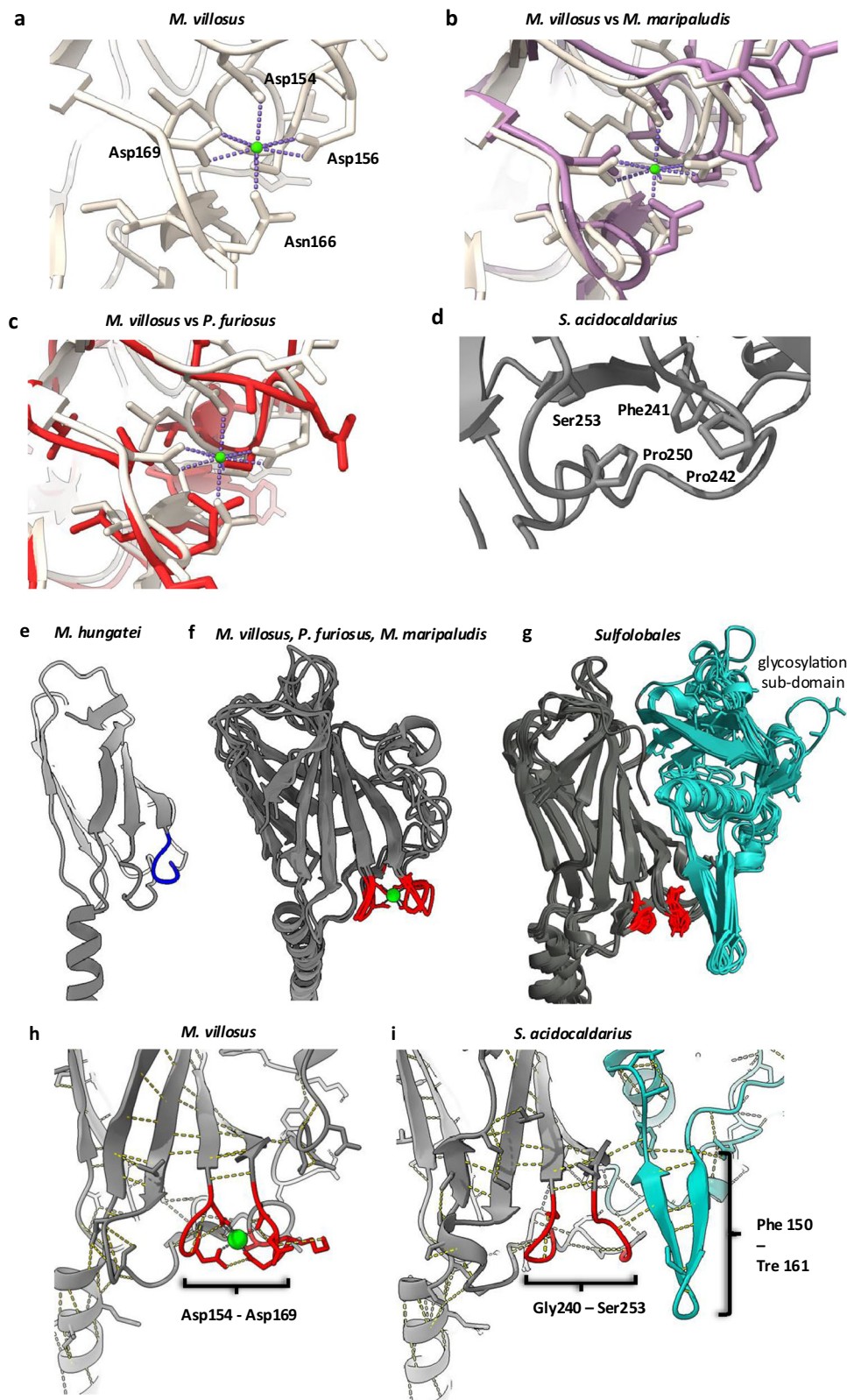

Moreover, archaella and Aap show different degrees of flexibility. While the Aap has the striking capability to bend up to 90°[9], archaella form stable supercoils necessary for swimming motion[57,61]. Interestingly, the *S. acidocaldarius* Aap show an unusual 3-conformer architecture, where each 3-start strand has a distinct conformation[9]. In *S. islandicus*, Aap even seem to exist in two states—one with a tri-

conformer architecture and another with a mono-conformer organisation. Compared to the tri-pilus, the mono-pilus borne subunits have a 'flapped' conformation, where the head domain has the opposite orientation with respect to the tail[41]. The 3-conformer structures observed in the Aap of *S. islandicus* and *S. acidocaldarius* have not been seen in any archaella to date, suggesting that it is unique to

**Fig. 3 | Archaellin head structures with and without ion coordination. a** Close-up of the metal binding site of *M. villosus* ArlB1. Superimposition of the metal site of *M. vollosus* (white) with those of *M. maripaludis* (**b**, purple) and *P. furiosus* (**c**, red). The ion is depicted as a green sphere. **d** Closeup of the corresponding, but ion-free site in *S. acidocaldarius*. **e** Close-up of the head domain of the archaellin from the euryarchaeon *M. hungatei* (PDB-5TFY[30]). The head domain is the smallest of all archaellin structures experimentally solved so far. There is no ion bound, as only a part of the coordinating loop is present (blue backbone segment).
**f** Superimposition of the archaellin head domains from the euryarchaea *M. villosus* (PDB-7OFQ[33]), *P. furiosus* (PDB-5O4U[31]) and *M. maripaludis* (PDB-5Z1L[32]). The head domains are larger than that of *M. hungatei* and contain a complete metal ion

coordination loop (coordinating residues in red). **g** superimposition of the experimental archaellin structures of *S. acidocaldarius, S. islandicus* (PDB-8CWM[67]) and Alphafold models of homologues listed in Supplementary Figs. 9 and 10. The glycosylation sub-domain that is conserved in Sulfolobales is coloured cyan. The region that overlaps with the ion coordination site in euryarchaeota is highlighted in red. In Sulfolobales, the residues in this loop are incompatible with ion binding. A β-hairpin (Phe150–Tre161 in *S. acidocaldarius*) of the glycosylation sub-domain packs against the ion-free loop, possibly compensating for the missing stabilising ion. Close-ups of the metal-coordination loop of *M. villosus* ArlB1 (**h**), and the corresponding ion-free loop and β-hairpin in *S. acidocaldarius* (**i**). Hydrogen bonds are shown as dashed yellow lines.

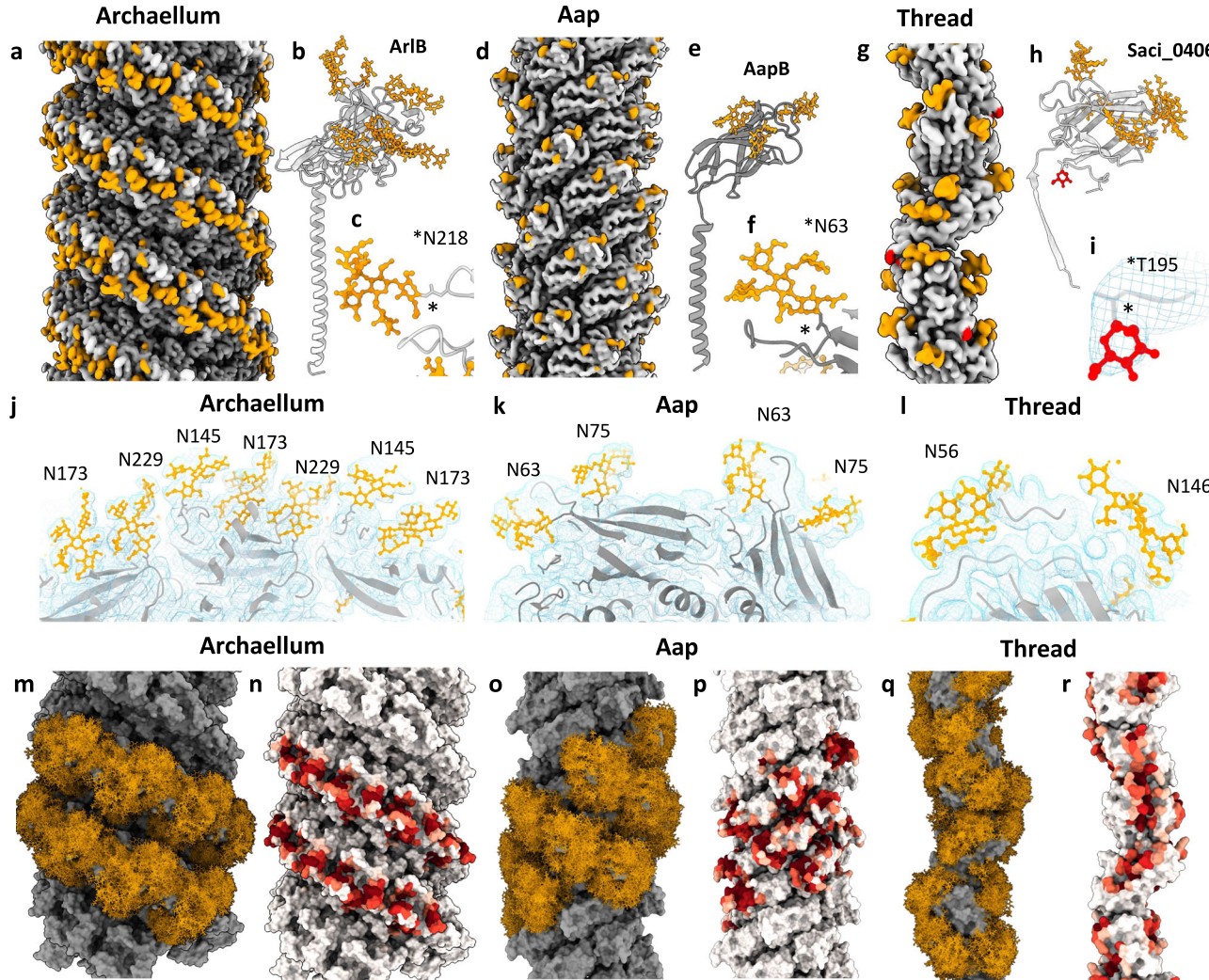

**Fig. 4 | The archaellum, Aap and thread of *S. acidocaldarius* in comparison.** CryoEM maps of the archaellum (**a**), Aap (**d**) and thread (**g**), atomic models of their subunits (**b, e, h**) and close-ups of selected glycans (**c, f, i**). The protein backbone in maps and models is coloured grey, N-glycans are orange and O-glycans red balls and sticks. Cross-sections through cryoEM maps with fitted models of the archaellum (**j**), Aap (**k**) and thread (**l**). Maps, transparent blue mesh; protein backbone, grey ribbons; glycans, orange balls and sticks. GlycoSHIELD[65] models

showing possible glycan conformers on the archaellum (**m**), Aap[9] (**o**) and thread[15] (**q**). Protein is depicted as grey surface and glycans as orange sticks. **n, p, r,** corresponding maps of surface shielding by glycans (ΔSASA, see 'Material and Methods'). ΔSASA scales are shown as white-to-red gradients with unmasked and most extensively shielded areas coloured in white and dark red, respectively. Inaccessible parts of the protein surface are shown in dark grey.

Aap. Considering that Aap drive twitching motility in *S. acidocaldarius* (and likely also in other Sulfolobales species)[12], it is conceivable that the 3-conformer structures are linked to this function, for example enabling rapid retraction[9].

On the other hand, Aap do not exhibit the screw-like profile that is caused by the glycosylation domain in the archaellum (Figs. 2d,

4d). This raises the question of whether these glycosylation ridges are linked to the function of the archaellum as a rotary propeller. In general terms, the diameter of a filament tends to be proportional to is stiffness[62]. The wider diameter of the archaellum may thus aid the formation of a stable superhelix necessary for swimming propulsion[57]. Wider archaella will be stiffer than narrower ones,

which in turn may influence the waveform of the superhelix. In resting state, archaella are also used as adhesive fibres[63]. The notched profile of the archaella may enhance the 'grip' of these archaella on solid surfaces.

Negative stain electron microscopy of cells harvested at different OD values shows that Aap are produced mainly in exponential phase, archaella are expressed then in late stationary phase or upon starvation (Supplementary Fig. 13)[42]. Aap filaments are formed within nutrient-rich conditions, thus aiding adhesion and biofilm formation. Conversely, archaella are up-regulated under starvation conditions, enabling the cell to travel to new locations in the search of resources[42,64]. This has also been observed for *S. islandicus*[41].

The biofilm-forming thread is composed of a beads-on-a-string arrangement of subunits, reminiscent of CU-pili (Fig. 4g, h, Supplementary Fig. 14). Contrasting with archaella, the threads are not confined to one cell pole and are expressed through all growth phases, suggesting that they act as 'baseline' adhesive filaments used under any condition[15]. Owing to its distinct architecture, the thread is considerably thinner (40 Å) than archaella (110 Å) and Aap (80 Å). Yet, in electron micrographs, it appears stiffer than Aap[15]. The apparent stiffness may be facilitated by two particularly strong types of intersubunit interactions; donor strand complementation (DSC) and isopeptide bonds[15]. These interactions also likely support the thread's tensile strength, which would greatly facilitate its proposed function in adhesion and biofilm formation. Our new 2.7 Å resolution cryoEM map of the thread reveals a putative O-glycan modification at residue T195 (Fig. 4i, Supplementary Fig. 14). It has previously been shown that O-glycosylation in *Sulfolobus* involves a single mannose molecule[52]. We were able to model a mannose molecule into the density, with α-mannose resulting in a better fit than β-mannose. The finding of the putative O-glycan is surprising, as so far, no O-glycosylation has been reported in any other experimentally solved surface structure of *S. acidocaldarius*. This O-glycosylation adds to the structural uniqueness of the thread, with currently unclear functional consequences.

Our new cryoEM maps of the archaellum, Aap and thread enabled us to build their complex N-glycans with greater precision (Fig. 4j–l). This was particularly the case for the Aap, where N-glycans were previously only partially resolved[9]. Glycans are usually more dynamic than proteins and rapidly explore a wide range of conformations[9,65]. As such, highly flexible glycans can cover most of the filament surface within hundreds of nanoseconds, as predicted by MD simulations. To visualise these glycoshields in the Aap and thread, we previously employed a molecular dynamics simulation approach (GlycoSHIELD[66]) and grafted all possible glycan conformers onto several neighbouring subunits (Fig. 4o–r[9,15]). Repeating this procedure with the archaellum filament (Fig. 4m, n), now allows us to compare all three filaments in terms of glycan mobility.

In the archaellum, all six glycan moieties are highly dynamic, likely aided by their exposed position on top of the glycosylation ridges. In contrast, any glycan positioned in the notches would experience steric hindrance by the surrounding protein domains, which would be entropically unfavourable, and restrict the archaellum's ability to bend and form supercoils. Indeed, it has been suggested that the degree of glycan dynamics determines the protein flexibility[9]. The *S. acidocaldarius* thread contains glycans that are wedged in between neighbouring subunits (Asn56 and Asn83) or into intramolecular clefts (Asn146) (Supplementary Fig. 14d). This restricts glycan flexibility (particularly that of Asn146), and thus also that of the filament[9]. In contrast, no such wedged glycans are present in the Aap pili, and it was proposed that this enables greater flexibility[9] (Fig. 4d–f, k, o, p).

### Glycosylation truncation does not alter the structure of motile filaments

We asked if the reduced swimming motility observed in glycan ablation mutants[40], is due to structural defects in *S. acidocaldraius'* motile

filaments—the archaellum and the Aap. To investigate this, we cultivated the *Δagl3* mutant. Negative stain TEM of *Δagl3* cells confirmed that the cell morphology was unaffected that archaella, Aap and threads were still produced (Supplementary Fig. 15). Moreover, we found that archaella extending from *Δagl3* cells still supercoil (Supplementary Fig. 16). Next, we purified archaella and Aap from this mutant (Supplementary Fig. 3 c, d) and solved their structures by cryoEM. The resulting maps had global resolutions of 2.4 Å for both filaments (Supplementary Figs. 4 and 5). In the maps of the mutant archaellum and Aap, the glycans were clearly altered. The densities for the sulfoqinuovose, the terminal glucose and one mannose were missing (Fig. 5a, b). However, comparing the protein structures of the archaella and Aap with and without glycan truncation revealed no significant differences (Fig. 5c–f, Supplementary Movies 1 and 2). These findings demonstrate that the motility-reducing effect of the glycan truncation is not due to structural changes in the archaella or Aap.

Interestingly however, only the O4-linked α-mannose was missing from the N-glycan trees of the *Δagl3* filaments, while the O6-linked α-mannose was still present (Fig. 5a, b). This suggests that the addition of the O4-linked α-mannose occurs downstream and is sensitive to the addition of sulfoquinovose. In contrast, the O6-linked α-mannose appears to be added independent of the presence of the sulfoquinovose. This indicates that the two mannose molecules are possibly added by different enzymes.

### Archaella, Aap and threads and their integration into the S-layer

We have recently determined an atomic model of the *S. acidocaldarius* S-layer[54], which is a hexagonal lattice consisting of the canopy-forming major S-layer protein SlaA, and trimers of SlaB anchoring the S-layer in the membrane. The S-layer contains alternating hexagonal and trimeric pores, thought to function as channels for various solutes and nutrients. However, due to a paucity of structural information from the same organism, it has so far not been possible to assess if S-layer pores could also act as conduits for motile or adhesive filaments. To address this question, we modelled archaella, Aap and threads through the pores of the *S. acidocaldarius* S-layer. This revealed that the threads are thin enough to traverse the hexagonal pores (Figs. 6a, 7a). In contrast, the diameters of the Aap and archaellum exceed that of either pore, meaning that a local reorganisation of the S-layer would have to take place to accommodate these filaments (Figs. 6b, c, 7b, c). For the archaellum, the removal of an S-layer hexamer delineating a hexagonal pore would create an aperture large enough for the archaellum to pass through the S-layer canopy (Figs. 6b, 7b). Due to the highly interwoven organisation of SlaA in the canopy, the removal of these subunits would not be detrimental to the structural integrity of the S-layer (Fig. 6b).

With a diameter of 80 Å (Fig. 6 c), Aap would either require a similar local disassembly of SlaA subunits, or at least a significant conformational change of the six SlaA N-termini delineating a hexagonal pore. The latter would require these N-termini to be pushed outward as the Aap grows through the pore (Fig. 7c). Indeed, single particle analysis of SlaA indicated that the N-terminal D1 domain is linked to the bulk of the protein (D2-D5) by a flexible hinge[54].

While further experimentation will be required to test these hypotheses, it is clear that the *S. acidocaldarius* S-layer has evolved to act as a semi-permeable barrier for the cell's surface filaments. In principle, this barrier would allow threads to be extruded anywhere, but would require local re-organisation for Aap and archaella. However, the mechanism for S-layer remodelling remains to be elucidated.

## Discussion

Solving the structure of the archaellum now completes our picture of the cell surface extensions of Saci under UV-stress free conditions. This enables us for the first time to compare three surface filaments from

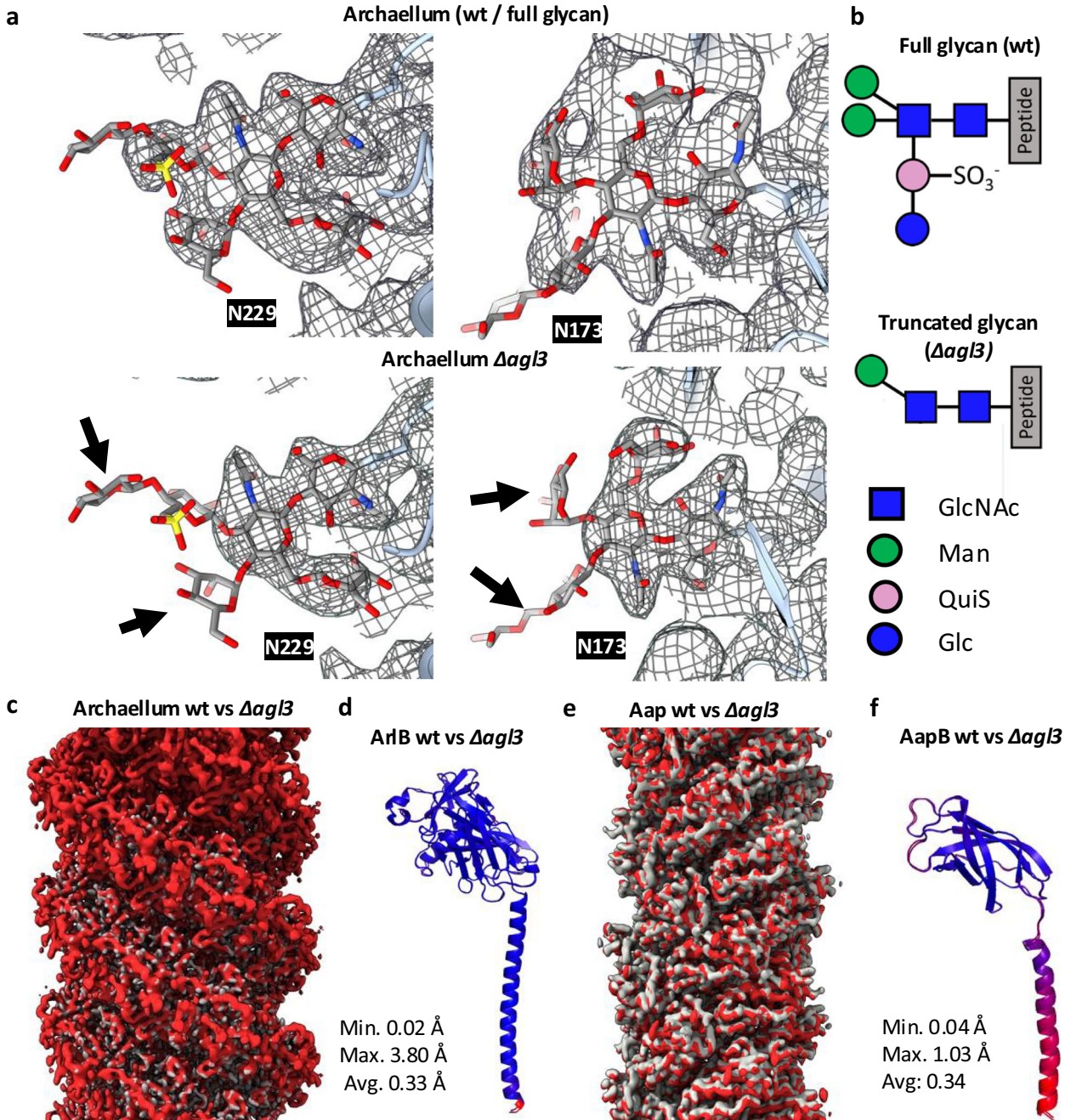

**Fig. 5 | Effect of glycan ablation on Aap and archaella of *S. acidocaldarius*.**
**a** (upper panel) Close ups of the cryoEM map (grey mesh) and model (ribbons and sticks) of the archaellum harbouring the full hexasaccharide glycan at positions N229 and N173. **a** (lower panel) Maps and models of the same positions in the *Δagl3* mutant, superimposed with the full-length atomic model of the glycan. The sulfo-quinovose, glucose and one mannose molecule of the model clearly lack corresponding densities (black arrows). **b** Schematic of the full N-glycan (top) and the truncated glycan of the *Δagl3* mutant (bottom). GlcNAc N-acetylglucosamine, Glc glucose, Man mannose, QuiS sulfoquinovose. Superimnposed CryoEM maps of the archaellum (**c**) and Aap (**e**). Maps of the archaellum and Aap with the full glycan are red, *Δagl3* mutants with the truncated glycan are grey. **d**, **f** Superimposed atomic models of wt and mutant archaellins coloured by RMSD (blue, low; red, high RMSD) show that glycan truncation does not alter the protein structure. Max, min and average RMSD values are indicated.

the same archaeal model species and shed new light on the question of how their distinct functions are reflected by unique structural features.

The differences in filament architecture between Aap and archaella reflect the mode of motility that these two filaments facilitate. Aap act as twitching ratchets requiring swift assembly and retraction. The Aap's 3-conformer organisation possibly represents a metastable state that could facilitate the retraction of the filament[9]. For archaeal Aap involved in twitching motility, this would be particularly important, as no retraction ATPase appears to be present[9,12]. The two Aap forms observed in *S. islandicus* (one tri-conformer pilus and another "flapped" mono-conformer pilus) may represent two retractive states, e.g. a pre-retractive and a post-retractive state[41].

In contrast, archaella propel cells though liquid media via rapid rotation. This requires a filament capable of adopting a stable super-helix when gyrated[57,61]. 

We find that the *S. acidocaldarius* archaellin ArlB archaellin contains a substantial glycosylation sub-domain that carries all six glycans.

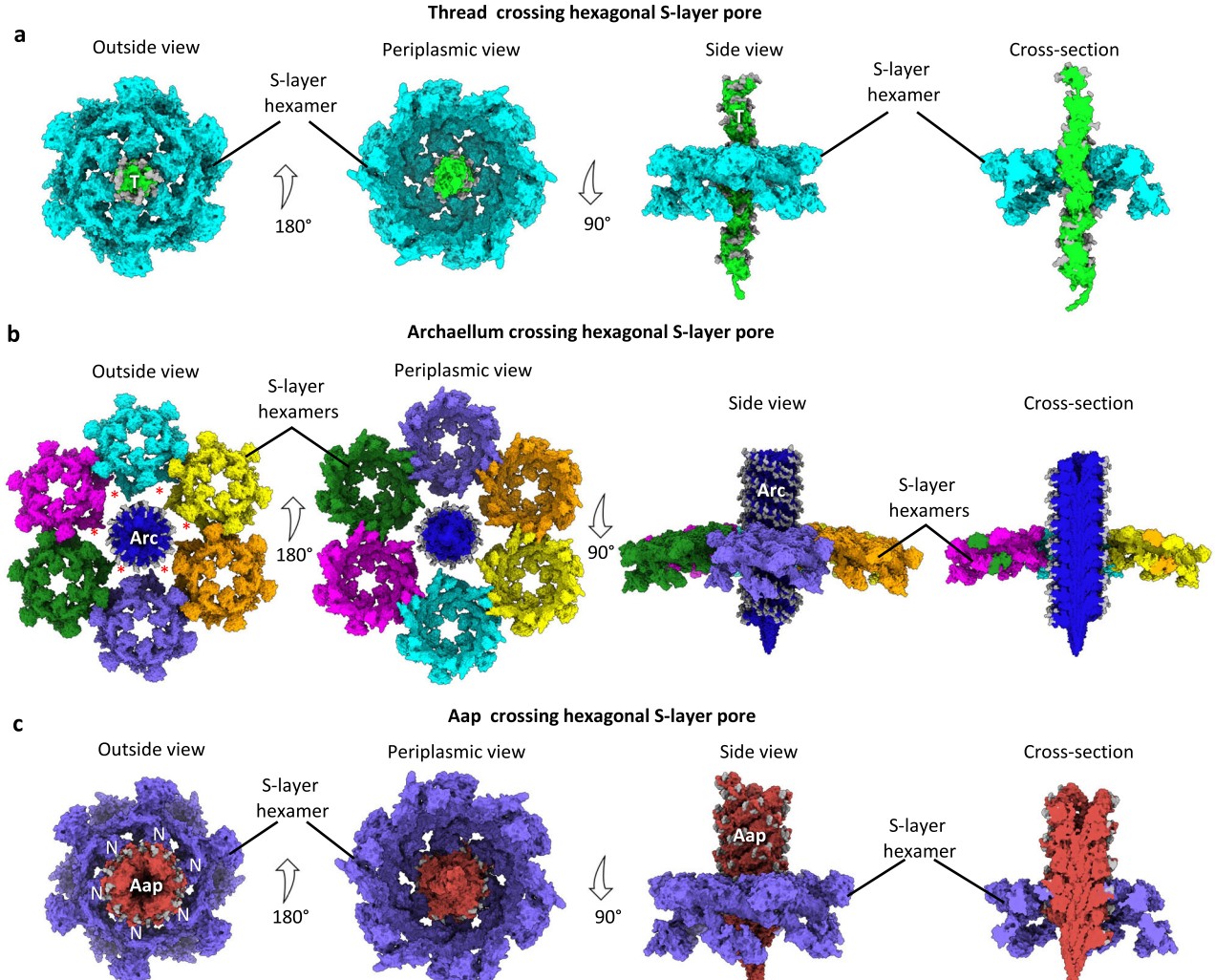

**Fig. 6 | Models of *S. acicocaldarius* filaments crossing the S-layer. a** Various views of one *S. acidocaldarius* S-layer hexamer (cyan, PDB- 8QPO[33]) with a thread (green) passing through it. The thread is sufficiently thin to fit through the pore. **b** Various views of the *S. acidocaldarius* S-layer (six SlaA hexamers shown in multicolour) with an archaellum (Arc, dark blue) passing through it. For the archaellum (Arc) to fit through the S-layer, the central SlaA hexamer would have to be removed (asterisks). **c** Various views of one *S. acidocaldarius* S-layer hexamer (purple) with an Aap (red) passing through it. The Aap filament clashes with the SlaA N-termini. The Aap could potentially fit through the pore, if SlaA N-termini (N) would fold outward. For simplicity, S-layer membrane anchors consisting of SlaB have been omitted.

This sub-domain is a widespread feature in crenarchaeal Sulfolobales species and gives their archaella a characteristic screw-like surface profile. Whereas the function and evolutionary advantage of these ridges remains to be revealed, it may be speculated that they modulate the stiffness of the archaellum and thus its superhelical waveform, which is essential for swimming motility. On the subunit level, the glycosylation sub-domain may also have a stabilising function in absence of a coordinated metal ion.

The threads are an entirely different class of filament and have evolved independently to the T4P[15]. Wedged glycans may limit their flexibility, meaning that the threads are stiffer compared to Aap and archaella. The threads are further reinforced by DSC and isopeptide bonds and are thus formidably strong adhesive fibres[15]. Threads form cables of multiple parallel or antiparallel filaments[15], and have been suggested to be involved in biofilm formation[67]. Moreover, threads resemble CU pili of Gram-positive bacteria[15,17,43], which are assembled in the periplasm in a chaperone-dependent manner and traverse the outer membrane through a membrane pore called an usher. Likely due to the lack of an outer membrane, a similar usher does not appear to be encoded in the thread operon[15]. Instead, the hexagonal pore of the S-layer may have evolved to adopt the role of the usher

and act as a scaffold that guides the thread into the extracellular medium.

In the thread, we also find a putative O-glycosylation site, unique to any of the surface structures of *S. acidocaldarius* investigated so far. It is possible that a currently elusive O-glycosyl transferase is distinctly associated with the thread assembly pathway and not with those of archaella and Aap.

In contrast to the threads, the S-layer does form a barrier to Aap and archaella, as neither of the two filaments fits through the S-layer pores without modification. Work by Umrekar et al.[28], showed that helices of ArlG and ArlF form between the cell membrane of *P. furiosus* and are predicted to bind to the S-layer. ArlF binds S-layer proteins in vitro[68] and ArlF and G have been proposed to coordinate the traversal of the archaellum through the periplasm[28]. Conceivably, these ArlFG filaments could be involved in remodelling the S-layer to allow the archaellum filament to pass.

Through solving the structures of archaella and Aap from the *Δagl3* mutant[40], we show that the structure of the filaments remains unaltered, despite the ablation of three terminal sugars, which includes the negatively charged 6-sulfoquinovose. However, *S. acidocaldarius Δagl3* mutants were virtually non-motile[40], indicating that

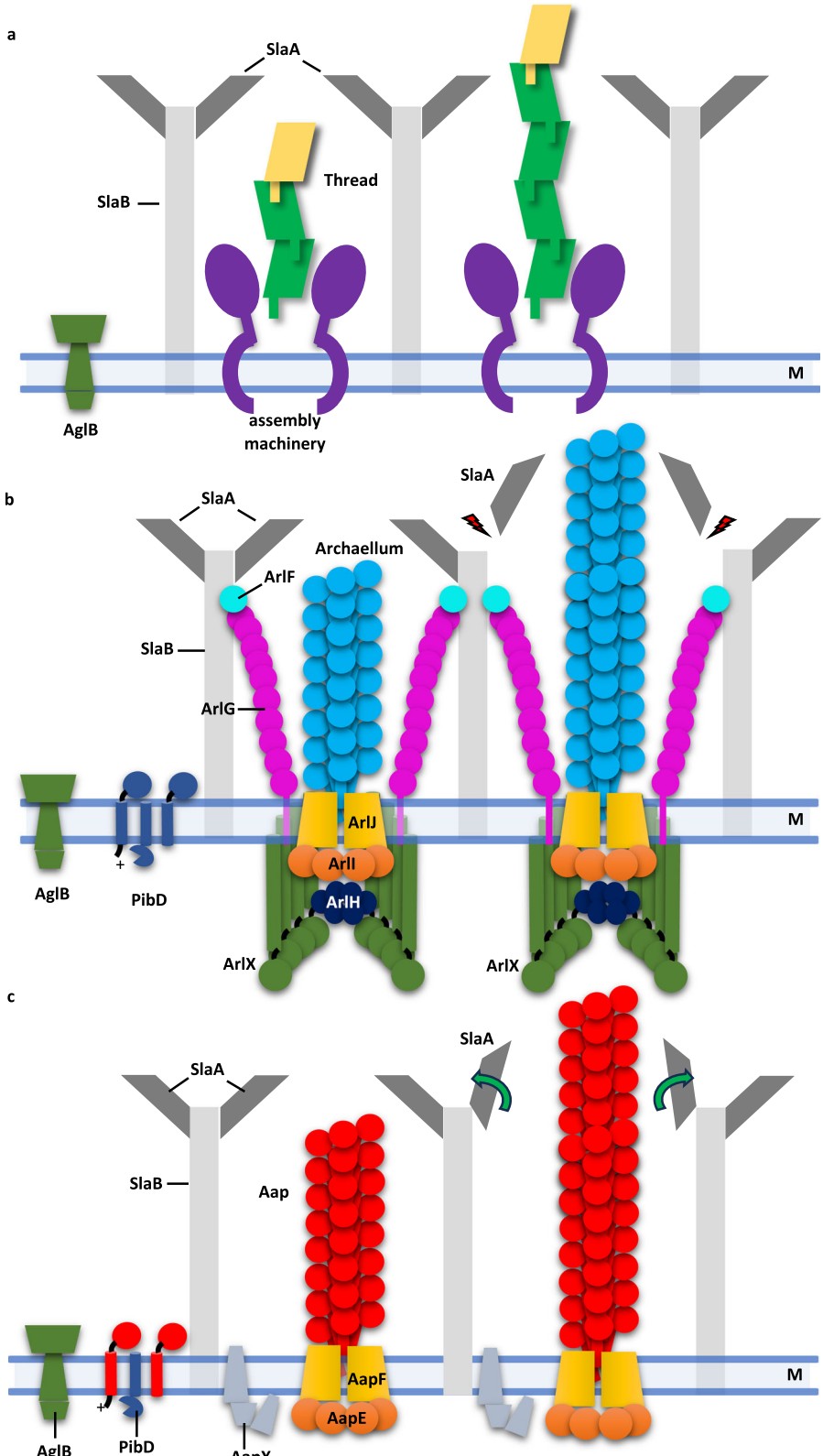

**Fig. 7 | Model of the interactions of archaella, Aap and threads with the *S. acidocaldarius* S-layer. a** Model of the thread assembly machinery[15]. The threads can pass though hexagonal pores of the S-layer. **b** Model of the archaellum assembly machinery[19,22,31]. ArlJ, assembly platform protein; ArlI, ATPase, ArlH, putative switch protein; ArlX, accessory protein with putative stator function; ArlG and ArlF anchor the machinery in the S-layer. A local reorganisation or disassembly of the SlaA pore would be required (red lightning symbol) for the archaellum to cross the S-layer. **c** Model of the Aap assembly machinery[9]. AapE ATPase, AapF assembly platform protein, AapX unknown function, M membrane. For the Aap filament to fit through the pores, SlaA subunits would have to be removed or at least fold outwards.

**Table 1 | Primers used for the generation of MW2106**

| Primer | Sequence (5′–3′) | Purpose |
|---|---|---|
| 11192 | TCCCAACGCGTACTGATGCCTTTATATCAGCGTTTTTCTCTAAAA | Δ *saci_1676-saci_1686* Upstream fw |
| 11193 | AGTATTAATAATTTAGTTGATACCTCTCTAATATCGAGATCCCTC | Δ *saci_1676-saci_1686* Upstream rev |
| 11194 | AGAGAGGTATCAACATAAATTATTAATACTTTTACACGAAAGTTC | Δ *saci_1676-saci_1686* Downstream rev |
| 11195 | ACCTAGGTCAGGATCGAATGGAGTGAAATATTAAGTA | Δ *saci_1676-saci_1686* Downstream fw |
| 11196 | TATTTCACTCCATTCGATCCTGACCTAGGTTTGAGCAGTTCTAGT | Linearization of pSVA407 |
| 11197 | CTGATATAAAGGCATCAGTACGCGTTGGGAGCTCTCCCATATGGT | Linearization of pSVA407 |
| 11144 | ACTGTATTAGGAGCAATAACACTAATTTCACTG | Sequencing primer binding downstream of *saci_1686* |

glycosylation is important for archaellum driven motility in *S. acid-ocaldarius*. It has been shown that single-point mutations of the N-glycosylation sites in ArlB had hardly any effect on motility and only the deletion of all 6 N-glycosylation sites resulted in a reduction of 40%[40].

Similar observations have recently been made in glycan truncation mutants of *Halobacterium salinarium*. Glycan ablation perturbed the swimming behaviour of the mutants compared to the wild type and increased bundling of archaella was observed[69]. The effect on bundling was not seen in *S. acidocaldarius* archaella or Aap, possibly because *S. acidocaldarus* has a smaller number of filaments per cell.

Nevertheless, it appears that rather than having major effects on the structure of individual filaments, N-glycans play a more important role in their dynamic behaviour and function. For archaella, one might speculate that the stator ArlF binds the S-layer via the N-glycans[68]. Once this interaction is abolished by shortening the glycan trees, torque generation may be impaired, similar to what has been observed in the S-layer knockouts[70]. Future research employing high-resolution imaging will be required to elucidate how the interaction between S-layers, filaments and surface glycans governs their assembly and function in motility, surface adhesion and biofilm formation.

## Methods

### Deletion of genes in *S. acidocaldarius*

The hyperarchaellated strain MW2106 was generated as follows. A deletion plasmid for *S. acidocaldarius* was constructed by amplifying up and downstream flanking regions (1000 bp) of the genes of interest, using the primers listed in Table 1. Overlap PCR was performed joining upstream and downstream fragments and cloned into pSVA407 containing the pyrEF cassette of *S. solfataricus*, resulting in the plasmid listed in Table 2. The plasmid was methylated by transformation into *E. coli* ER 1821 containing pM.EsaBC4I80. Transformation of pSVA12926 into *S. acidocaldarius* and generation of deletion mutant was done as described previously[71]. The deletion was confirmed by PCR and DNA sequencing. The deletion strain is listed in Table 3.

### *S. acidocaldarius* cultivation and archaellaum isolation

*S. acidocaldarius* MW2106 and MW039 (Δ*agl3*)[40], were inoculated from cryo-stock into 50 ml basal Brock medium (at pH 3.5) supplemented with 0.2% (w/v) Dextrin, 0.1% (w/v) NZ-amine and 10 μg/ml uracil. These precultures were grown for three days at 75 °C with light agitation. Aliquots of precultures were inoculated into freshly prepared 2 × 1 l Brock media (at pH 3.5) and cells were grown until late stationary phase. The cells were harvested at 4700 × *g* for 25 min at 4 °C and the pellets were resuspended in 20 ml of Basal Brock medium without FeCl₃ (at pH 3.5). The filaments were sheared using a peristaltic pump (Ismatec) which was connected to a syringe needle with 1.1 mm diameter and 25 mm length (BD Microlance). The cells were pumped through the syringe for 45 min at 25 rpm speed. The syringe needle was exchanged for a narrower one with a diameter of 0.5 mm and length of 25 mm (BD Microlance) and the cells were pumped through the

**Table 2 | Plasmid used for the generation of MW2106 Plasmid**

| | | |
|---|---|---|
| pSVA12926 | In-frame deletion of *saci_1676-saci_1686* | This study |

**Table 3 | Strains used**

| Strains | Genotype | Source/reference |
|---|---|---|
| MW2106 | Deletion of *saci_1676-saci_1686* in MW001 | This study |
| MW039 | Deletion of *agl3* in MW001 | Meyer et al.[40] |

syringe for 1 h at 25 rpm speed. After shearing, the cells were centrifuged at 12,000 × *g* for 25 min at 4 °C. The supernatant was carefully collected and centrifuged at 200,000 × *g* for 1.5 h at 4 °C. The resulting pellet was resuspended in 500 μl Basal Brock without FeCl₃ and subjected to density gradient centrifugation. The resuspended pellet was layered on 1.5 ml CsCl (0.5 g/ml) and centrifuged at 250,000 × *g* for 16 h at 4 °C. Following this step, a white band in the upper half of the tube was collected. It was diluted to a volume of 10 ml in Basal Brock media without FeCl₃ and pelleted at 250,000 × *g* for 1 h at 4 °C. The pellet was resuspended in 50 μl Basal Brock media without FeCl₃ and stored at 4 °C. All the components for making Brock media solutions were purchased from Sigma-Aldrich, UK. Oligonucleotides were obtained from IDT (USA), Dextrin from Carl Roth (Germany) and Gelrite from Duchefa Biochemie (The Netherlands).

### Negative stain transmission electron microscopy

5 μl *S. acidocaldarius* cells of both strains were applied to freshly glow-discharged 300 mesh carbon-coated copper grids (Plano GmbH) then incubated for 30 s. The excess liquid was blotted away, and Milli-Q water was applied to the grid before being blotted off as a wash step. This was then repeated three times before grids were stained with 2% Uranyl acetate. A Thermo Fisher Tecnai Spirit (Thermo Fisher Scientific) operating at 120 kV and equipped with a Gatan OneView detector, or a Hitachi HT8600 operating at 100 kV and equipped with an EMSIS XAROSA camera (EMSIS) were used for imaging.

### Cryo-EM sample preparation and data collection

3 μl of concentrated filament solutions of both strains were pipetted onto glow-discharged 300 mesh copper R2/2 Quantifoil grids. 597 Whatman filter papers were utilised to blot the grid for 5 s, combined with a blot force of 1 and in an environment of 95 % relative humidity and 21 °C. Plunge-freezing of the samples into liquid ethane was performed using a Mark IV Vitrobot (Thermo Fisher Scientific). A 120 kV FEI Tecnai Spirit EM (Thermo Fisher Scientific), combined with a Gatan OneView CMOS detector was used to screen the grids.

For strain MW2106, an initial data collection was performed using a Talos Arctica TEM (Thermo Fisher Scientific) at an accelerating voltage of 200 kV with a K2 direct electron detector (Gatan/Ametek). A magnification of ×130,000 was used, equating to a calibrated pixel size

of 1.05 Å. A total of 3234 movies were recorded at a dose rate of 8.21 e/Å$^2$ s$^{-1}$ with 44 fractions, an exposure time of 5.5 s and a total dose of 45.2 e/Å$^2$ s$^{-1}$. A defocus range of −0.8 to −2.0 μm was used, with 0.4 μm steps.

A Titan Krios G3 Cryo-TEM (Thermo Fisher Scientific) was used to collect high-resolution image data for all data sets. The Krios was operated in nanoprobe mode, using parallel illumination and coma-free alignment at an acceleration voltage of 300 kV. This was paired with a K3 BioQuantum direct electron detector and GIF energy filter (Gatan/Ametek). Both detectors were operated in counting mode at calibrated magnifications of ×105,000 (relating to a pixel size value of 0.829 Å) and ×130,000 (relating to a pixel size value of 0.921 Å) for strains MW2106 and MW039, respectively. EPU software (Thermo Fisher Scientific) was used in both instances to control the data collection.

Movies of archaella filaments from strain MW2106 were recorded at a dose rate of 27.91 e/Å$^2$ s$^{-1}$ with 43 fractions, 1.55 s exposure, with an accumulated total dose of 43.27 e/Å$^2$ and a set defocus range of −1.0 to −2.5 μm, using 0.3 μm steps. Movies for archaella and Aap from strain MW039 (Δagl3) were recorded at a dose rate of 8.8 e/Å$^2$ s$^{-1}$, in EER format, 4.5 s exposure and a total accumulated dose of 40 e/Å$^2$. A defocus range of −0.8 μm to −2.2 μm with 0.2 μm steps was used. EER movies were split into 40 fractions with a 2x upsampling factor (8k × 8k). Further details are listed in the supplementary cryo-EM statistics tables.

## Cryo-EM image processing

Data processing for both strains was performed using cryoSPARC[55]. A total of 20,579 and 29,189 movies were collected for strains MW2106 (archaella with full-length glycans) and MW039 (Δagl3; archaella and Aap with the truncated glycan), respectively. Patch motion correction and CTF estimation were performed for all movies. For archaella of strain MW2106, a subset of micrographs was used for manual picking to provide templates for filament tracing across the whole dataset. A total of 4,748,436 helical segments were picked with an inter-particle spacing of 40 Å, which were then subjected to multiple rounds of 2D classification to remove false positives and low-quality particles. 1,082,952 particles were selected and used for further processing. Initial helical refinements with no parameters were performed, from which rough helical parameters were determined visually in ChimeraX[72], and then checked using cryoSPARC's symmetry search tool[55]. Multiple rounds of helical refinement, CTF refinement and reference-based motion correction were performed until no improvements were seen in resolution. The final reported resolution of the archaellum map was 2.0 Å, using 1,059,736 helical segments for the reconstruction. The helical rise and twist values used for reconstructing the archaellum were 5.433 Å and 107.923°, respectively. The box size of the final refinement was 288 and a helical symmetry order of 7 was applied.

For the archaella of strain MW039 (Δagl3), the filament tracer was used for particle picking on a subset of 1927 micrographs, using the same inter-particle spacing and manually picked helical segments mentioned previously as reference templates. From this, a total of 1,174,572 particles were selected. After iterative 2D classification, 322,074 particles were used for the initial helical refinement, using the same parameters that were determined for strain MW2106. The data were subjected to multiple rounds of CTF refinement and reference-based motion correction. A final helical refinement resulted in an average resolution of 2.4 Å using 256,869 particles. Helical rise and twist values were determined as 5.399 Å and 107.937°, respectively. Again, the box size used for the final refinement was 288 and a helical symmetry order of 7 was applied.

Aap filaments of an *S. acidocaldarius* strain MW158 (lacking archaella and UV-pili) were reprocessed from a dataset collected previously[32]. A total of 7,833,953 helical segments were picked from 6272 micrographs using the filament tracer and then subjected to 2D

classification. 505,862 particles were selected and used for helical refinement. The approximate helical parameters of 15.5 Å rise and 40° twist from our previous study were used as a starting point. As for the archaella, multiple rounds of CTF refinement and reference-based motion correction were performed. A final average resolution of 2.6 Å was achieved with a box size of 288 and a helical symmetry order of 3. Optimised helical rise and twist values used for the reconstruction were 15.524 Å and 39.859°.

The Aap from strain MW039 (Δagl3) were picked from 27,362 micrographs using the filament tracer. 2D class averages from the original MW2106 processing were used as templates. A total of 3,560,700 helical segments were picked and then subjected to 2D classification. 692,629 particles were selected and used for helical refinement. Again, multiple rounds of CTF refinement and reference-based motion correction were performed. A final average resolution of 2.4 Å was achieved using 691,479 particles. Optimised helical rise and twist values used for the reconstruction were 15.282 Å and 39.953°, with the same box size and helical symmetry order as with the MW158 strain.

The threads with the full-length glycan from strain MW2106 were picked from a subset of 10,059 micrographs. A total of 4,735,170 segments were picked with an inter-particle spacing of 63 Å and subjected to multiple rounds of classification. Initial helical refinement was performed using a selection of 451,248 particles. Initial rise and twist values of 31.6 Å and −103° as determined in our previous study[15]. After two rounds of CTF refinement and reference-based motion correction, a final helical refinement was performed using all the available particles, a total of 626,078 particles. An average resolution of 2.7 Å was achieved, with helical rise and twist values determined to be 31.650 Å and −103.281°, respectively. The box size and helical symmetry order for the final reconstruction were 288 and 2, respectively.

## Model building and validation

The resolution of 2.0 Å for the MW2106 archaellum enabled unambiguous model building in Coot[73], to identify ArlB as the major archaellin subunit. Glycosylation patterns within ArlB were also seen. The model was further corroborated through comparison between our modelled ArlB and AlphaFold2[59] predictions. MOLREP[74] then allowed for phased molecular replacement to position the remaining monomers into the density. CCP4[75] allowed changes in the rebuilt model to be propagated, to accurately fit the copied monomers into the other subunit densities. Glycan modelling was performed using Coot[73]. For this, we prepared a dictionary of unusual sugars using JLIGAND[76]. Final refinement of the finished structure was performed with REFMAC5[77], via the CCPEM interface[78].

Models of Archaellum, Aap and thread glycan truncation (Δagl3) mutants were built by molecular replacement of the MW2106 archaellum, and previously published structures of "wild type" Aap[9] and threads[15], respectively. Molecular replacement was also performed to build the updated models for the wild-type Aap and thread. All atomic models were refined by REFMAC5[77] as above. CryoEM and model building statistics are summarised in Supplementary Tables 1 and 2.

## Sequence analysis and structural prediction

Homology searches for other major archaellin homologues within related crenarcheota and euryarchaeota species were performed using SyntTax[58]. The most similar results were then compared against each other using Clustal Omega[79]. The KEGG genome database[80] allowed for visualisation of the predicted gene clusters surrounding ArlB in *S. acidocaldarius,* as well as related species. Protein structure prediction was performed with AlphaFold2, using the online ColabFold tool[59] and ConSurf[81], which enabled the structural similarity of the ArlB homologues to be ascertained. The AlphaFold2 predications have been provided as supplementary dataset 1.

## Glycoshield analysis

Arrays of glycan conformers were grafted onto the protein structure using GlycoSHIELD[65,82]. Briefly, glycan systems (GlcNAc[2], Man[2], QuiS[1], Glc[1] N-linked to neutralised glyc-Asp-gly tripeptides) were simulated in CHARMM-GUI[83], solvated with TIP3P water models supplemented with 150 mM NaCl, and configured for simulation using CHARMM36m force fields[84,85]. Molecular Dynamics Simulations (MDS) were carried out using GROMACS 2020.2 and 2020.4-cuda100 in hybrid GPU/CPU setups. Initial energy minimisation (5000 steps, steepest descent algorithm) was followed by equilibration in the NVT ensemble (1 fs time-steps, Nose-Hoover thermostat). Restraints were applied to atom positions and dihedral angles during equilibration, with initial force constants set at 400, 40, and 4 kJ/mol/nm$^2$ for backbone positions, side chain positions, and dihedral angles, respectively. These force constants were gradually reduced to 0. Systems underwent additional equilibration in the NPT ensemble (Parrinello-Rahman pressure coupling, time constant of 5 ps, compressibility of $4.5 \times 10^{-5}$ bar$^{-1}$) over 10 ns with a time step of 2 fs. Hydrogen bonds were constrained using the LINCS algorithm. In production runs, a velocity-rescale thermostat maintained the temperature at 351 K, with a total duration of 3 μs, and snapshots of atom positions were stored at 100 ps intervals.

The resulting glycan conformers were grafted onto protein structure with GlycoSHIELD using a threshold distance of 3.25 Å between protein α-carbons and glycan ring-oxygens. Following shuffling and subsampling, glycan conformers were selected to represent plausible structures for visualisation. Molecular renders were generated using ChimeraX[72].

## Reporting summary

Further information on research design is available in the Nature Portfolio Reporting Summary linked to this article.

## Data availability

The cryoEM maps generated in this study have been deposited in the EM DataResource and the Protein Databank under the following accession codes. EMD-18700/PDB 8QX4 for the wild type archaellum, EMD-19608/PDB 8RZL for the wild type thread, EMD-19960/PDB 9ETS for the wild type Aap, EMD-19961/PDB 9ETT for the Δagl3 archaellum, and EMD-19990/PDB 9EV0 for the Δagl3 Aap. Previously published structures used for comparison can be found in the PDB databank (https://www.rcsb.org) under the following accession codes: PDB-8CWM for the *S. islandicus* archaellum[58]; PDB-7OFQ for the *M. villosus* archaellum[33]; PDB-5Z1L for the *M. maripaludis* archaellum[32]; PDB-5O4U for the *P. furiosus* archaellum[31]; PDB-5TFY for the *M. hungatei* archaellum[30]. The raw image data used in this study have been deposited to the Electron Microscopy Public Image Archive (EMPIAR) under accession numbers EMPIAR-12184 for the wild-type filaments isolated from strain MW2106 and EMPIAR-12196 for the Δagl3 mutant filaments from strain MW039. The *S. acidocaldarius* (DSM639) genome can be accessed via the KEGG accession code T00251 or the NCBI gene bank code CP000077 112. The transcriptomics data analysed in this study can be accessed in the Pan Genomic Database for Genomic Elements Toxic To Bacteria under the following link: https://exploration.weizmann.ac.il/TCOL/index_singleOrg.php?organism=sulfolobus_acidocaldarius&tab=0.

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

## Acknowledgements

We thank Diamond Light Source for access to the cryoEM facilities at the UK national electron bio-imaging centre (eBIC), funded by the Wellcome Trust, MRC and BBSRC. eBIC access was granted under the BAG allocations BI25452 and BI32707. We acknowledge access and support at the GW4 Facility for High-Resolution Electron Cryo-Microscopy, funded by the Wellcome Trust (202904/Z/16/Z and 206181/Z/17/Z) and BBSRC (BB/R000484/1). We are grateful to U. Borucu of the GW4 Regional Facility for High-Resolution Electron Cryo-Microscopy for help with screening. B.D., M.G., M.M. and R.U.H. were supported by an ERC Starting Grant under the European Union's Horizon 2020 research and innovation programme (grant agreement No 803894), awarded to B.D. M.M. was also funded by a BBSRC New Investigator Research Grant (BB/R008639/1) and Leverhulme Trust Project Grant (RPG-2023-069), both awarded to V.G. S.S. and S.V.A. were supported by the Collaborative Research Centre SFB1381 funded by the Deutsche Forschungsgemeinschaft (DFG, German Research Foundation)—Project-ID 403222702—SFB 1381. S.S. and S.V.A. were also funded by the Deutsche Forschungsgemeinschaft (DFG, German Research Foundation) under Germany's Excellence Strategy (CIBSS – EXC-2189 – Project ID 390939984). C.H. was supported by the Agence Nationale de la Recherche (grants #ANR-16-CE16-0009-01 and #ANR-21-CE16-0021-01). For the purpose of open access, the author has applied a 'Creative Commons Attribution (CC BY) licence to any Author Accepted Manuscript version arising from this submission.

## Author contributions

Major contributions to (i) the concept or design of the study (S.A., B.D.,) (ii) the acquisition, analysis, or interpretation of the data (M.G., M.I., M.M., C.M., J.S., S.S., R.H., C.H., D.K., B.D.); (iii) writing of the manuscript (M.G., M.I., C.M., S.S., R.H., V.G., S.A., B.D.) and provision of resources (B.D., V.G., S.A.).

## Competing interests

The authors declare no competing interests.
