## [Transparent Peer Review file · Nature Communications]

Towards a molecular picture of the archaeal cell surface

Corresponding Author: Dr Bertram Daum

Version 0:

Reviewer comments:

Reviewer #1

(Remarks to the Author)

In the manuscript "Towards piecing together a molecular picture of the archaeal cell envelope" Gaines et al. report the structures of several surface filaments from the crenarchaeon *Sulfolobus acidocaldarius* with an emphasis on the surface glycans attached to these filaments and how the filaments protrude the S-layer. The main new finding described in this manuscript is the structure of the archaellum and its "glycosylation domain". In addition to the archaellum structure, this manuscript contains a lot of data, some of which is not convincing, and some does not seem related to the main finding, such that the manuscript reads as a lot of pieces stitched together rather than one comprehensive story.

Although reported for the first time for this organism, structures of archaella from several species have already been determined, and a subdomain rich with glycans was also seen in *S. islandicus* (Kreutzberger et al., 2022). The significance of the work as it is presented is not clear, and the work should be revised to support claims made in this manuscript (or omit these claims).

Specific comments:

1. The authors use the MW2106 strain to obtain archaella. What is the mechanism of hyper-archaellation in the MW2106 strain? Could it affect filament architecture? There is no reference to the work in which this strain was generated, only a mention in Table 1 that it is from the Albers lab strain collection. If the generation of this strain was not reported earlier, its generation should be described in the Methods and further characterization of this strain should be presented, e.g., evidence that it is hyper-archaellated and that other aspects of physiology were not severely compromised.
2. Cryo-EM data: a. When referring to resolution, please state that this is an average resolution of the map, as these are cryo-EM maps. b. Images of cryo-EM maps are cropped. Full images of maps with estimation of local resolution should be displayed, at least in the supplementary. c. Sample cryo-EM micrographs should be displayed in the supplementary. d. Box sizes, overlap between particles, initial helical rise and twist used in the reconstruction (not just optimized in the final reconstruction) should be mentioned in the methods for each reconstruction.
3. The authors report updated structures of the Aap and thread. Improving resolution is always nice, but the insights gained here are not clear. For the Aap, no new finding is reported. For the thread, the authors write that they discovered a so-far unknown O-glycosylation site. How do they know that the density near T195 is a glycan and not some other modification? Could a phosphate or sulphate group fit in there, for example? Density in the cryo-EM map alone cannot determine the identity of the modification, and additional data (e.g., mass spectrometry) should be provided to support this claim.
4. Comparison between wild-type and Δ agl3 filaments: a. The motivation for this comparison is not clear at all. The first time that glycosylation is mentioned in the introduction is in line 99: "To tackle the long standing question of how glycans impact on filament architecture...". More information on glycosylation in archaea is necessary, at the very least, a mention of the extent of surface glycosylation in archaea and why glycans might impact filament architecture. Mentioning the types of glycosylation found so far in *S. acidocaldarius* would also be helpful. b. "Comparing these structures with those previously published for strains expressing full-length glycans 33,38 revealed no significant differences... (Figure 4)." Please be more quantitative. What is a significant (or not significant) difference? Can one calculate the RMSD between structures in certain regions? Figure 4 does not show this point convincingly. Superimposing models would be better. c. The authors mislead the reader by writing that they solved the structure of the thread from the mutant (abstract, results, discussion). In the Methods they write that they built a model of the thread from the mutant. However, a model was not built (=structure was not solved), as revealed by the authors after I requested the model, map, and validation report for this structure. Moreover, the authors report a 4.1 Å resolution for this map, but when receiving the map and looking at it, it is clear that this is an overestimation. The quality of the map is poor. It is blobby, with no secondary structure observed, while

secondary structure and some side-chain density should be visible at the reported resolution. With this kind of data, I do not understand how the authors conclude that the structure of the thread is unaffected by the lack of the Sulfoquinovose (line 221). The authors should either improve the map, build a model, and make a proper comparison, or omit this entire part from the manuscript.

5. Metal-binding site: a. The text refers to this issue after presenting figs. 3 and 4, yet the data is presented in fig. 2c. This is very confusing. I recommend moving this figure to the supplementary. b. Fig. 2c is tiny and it is extremely hard to see the metal ion depiction. c. "Revisiting the archaella structures from *P. furiosus* and *M. hungatei*, we find that the *P. furiosus* archaellum also contains the same metal coordination site". Revisiting means the authors inspected again the cryo-EM map? If so, the density of the metal ion should be shown. d. Hydrogen bond numbers are reported for different structures. How was this calculation done? Is the calculation normalized to the number of residues? In *S. acidocaldarius* there are more residues, thus the number of hydrogen bonds is expected to be higher. So, is the argument here that larger archaellins are more stable than smaller ones?
6. Some figures are not clear and hardly convey the message written in the text/caption. Examples: 1. Supplementary Figs. 3 and 9, and especially Fig. 4d, are hard to interpret. Why is the mesh in Fig. 4d shown differently for wt and the mutant (more dense and smooth for Δ agl3)? I recommend making the mesh partially transparent, so that it is easier to see the model, and to remove from the figure regions of the map and model that are irrelevant to what the authors are trying to show. 2. "The *S. acidocaldarius* thread contains a glycan linked to N146, which is wedged in between neighbouring subunits, thus likely restricting both the flexibility of the glycan as well as that of the filament 33 (Figure 9 c)." Should be supplementary Figure 9c. Also, this point is not clear at all from this figure. It would be helpful to color two neighboring subunits differently and show that the glycan is wedged between them.
7. Since Nature Communications is not a specialized journal for archaea, please explain the term "crenarchaeota".
8. The first paragraph of the introduction contains a lot of details on archaella components, which is hard to follow. Is all this information necessary? If so, an illustration in the supplementary would help the reader to follow.
9. In the introduction, the authors explain that several archaellins may be encoded, depending on the species. However, there is no mention of how many, and which are encoded in *Sulfolobus acidocaldarius*. Later, they conclude that ArlB is the only component of the archaellum based on the cryo-EM map. Are there any other archaellins that did not fit the map?
10. Fig. 2b: Doesn't look like all images are on the same scale (For example, the purple one looks bigger than others).
11. The captions of supplementary Figs. 7,8 report a scale bar, but I cannot find a scale bar in these figures.
12. "Interestingly, only the O4-linked α -mannose is missing from the glycan tree, while the O6-linked α -mannose is still present... This may indicate that the two mannose molecules are added by different enzymes." Why is this interesting (as stated in the beginning) or new? Wasn't this already resolved in Meyer et al., 2011? The structure is just consistent with the earlier finding.
13. There are two instances where the authors write that they investigated AlphaFold2 prediction models, yet no figure/table/text shows this data (metal binding site and glycosylation domain).
14. Supplementary Fig. 11: a. The caption states: "xxxx strain". b. It is not clear from the caption what the difference is between a and b, other than the magnification. c and d. Based on what were these maps/models placed next to each other at these positions? It is impossible to see from these views that ridges and notches interlock and that residues do not clash with each other.
15. Fig. 7: the caption states that b shows the archaellum and c the Aap, but the figure shows the opposite.
16. In the discussion: "While archaella have been seen to bundle in multiarchaellated archaea 66–68, *S. acidocaldarius* has few archaella per cell and archaella bundling is not observed in swimming motion." The authors cite a paper in which archaella bundling is observed in swimming motion of *Halobacterium salinarum*. Is there a paper showing directly, with a comparable technique, that the archaella of *S. acidocaldarius* do not bundle in swimming motion? If so, it should be cited. If not, the authors should refrain from making such a statement.
17. The presentation could be improved. There are several typos, changes in font in the supplementary, inconsistency in naming the mutant strain (Δ agl, Δ agl, Δ agl), etc.

Reviewer #2

(Remarks to the Author)

In this article, Gaines et al provided an in-depth structural analysis of the protein filaments from *Sulfolobus acidocaldarius*, an archaeal species known for its complex motility and adhesion mechanisms. Using cryo-electron microscopy, the study achieves high-resolution structures of the archaellum and other associated filaments, including novel insights into their N-glycan modifications and architectural dynamics. These findings deepen our understanding of archaeal cell envelope structures and their functional implications, highlighting distinct glycosylation patterns that influence filament stability and motility. Overall, the story is good, and my co-reviewer and I have some issues that need to be addressed in a revision.

Major issues:

- (1) The authors used the hyper-archellated strain MW2106 to express high amounts of archaella for cryoEM. Are they able to collect electron cryo-tomography data of one of these cells? This could be really cool if they are able to visualize emerging filaments and relate their position to the S-layer lattice.
- (2) The N-glycosylation observation is great. Could the authors provide figures that adjust the map threshold to clearly show the extra glycan densities that come from Asn but not adjacent S/T? This would be very helpful for the readers to immediately understand how you know it's N-linked glycosylation.
- (3) The authors mention that "*S. acidocaldarius* glycans are tri-branched hexasaccharides." Given that N-linked

glycosylation is generally better understood than O-linked glycosylation, it would be beneficial for the authors to use mass spectrometry to validate the components of these glycans. If this is not feasible, a rationale for its impracticality should be provided. Additionally, in Fig. 4d, where glycans are included in the model, the authors need to clarify whether these are merely illustrative or if they were actually incorporated into the structural model. If it's the latter, supporting mass spectrometry data should be included to validate the glycan structures depicted.

(4) The impact of the Δ Agl3 mutation on motility is intriguing. However, it remains unclear whether this mutation significantly affects the morphology of the flagella, such as supercoiling or the number of filaments produced. To clarify this, the authors should include negative staining images that compare the morphology of the flagella from both wild-type and mutant strains. This would help determine if the observed changes in motility correlate with structural alterations in the flagella.

(5) I noticed that the term "archaellum" is also referred to as "archaeal flagellum" in some other papers. While I personally have no preference for the terminology used, it can be confusing for readers unfamiliar with this field. After randomly asking several colleagues who do not specialize in archaea or flagella-related topics, it appears that "archaeal flagella" is more easily understood. Although it is too late to change the terminology in the field, could the authors clarify the first mention of "archaellum" to indicate that it is also known as "archaeal flagellum"? This small addition could help improve clarity for a broader audience.

Minor issues:

Line 330, is there a reference supporting the described relationship between filament dimensions and mechanical forces?

Line 336-341, Author's have not mentioned here that Aap in *S. islandicus* has been shown to exist in two conformations, one which has a tri-conformer arrangement and the other with a mono-conformer. These two conformers have opposite orientations of the C-terminal globular domain with respect to the N-terminal helical domain. In light of the author's suggestion that the 3-conformer could support rapid retraction, might they mention these two distinct forms? This evidence from *S. islandicus* seems relevant to discussion of rapid retraction if, for instance, the two states are conformational exchangeable (one representing pre-retraction, the mono-conformer, and the other representing active retraction or post-pre-retraction).

Structure Table. Why is sharpening B factors is 0? What is Model Refinement resolution? I've never heard this term before. Model resolution: model is atom coordinates, and it doesn't have a resolution. You should report this as model: map FSC (0.5) and put it under "map resolution estimation".

I didn't see any raw micrographs (cryo-EM, not negative staining) or 2D averages showing what those three different filaments look like. This is helpful information to include somewhere in the paper.

Reviewer #3

(Remarks to the Author)

Reviewer #4

(Remarks to the Author)

Gaines et al present 3 types of archaeal filaments from *S. acidocaldarius* at resolutions up to 2 Angstroms – the highest achieved to date for these archaeella, Aap and threads. This enabled the authors to identify and model glycans in all three filaments, identifying a novel γ -linked glycosylation site. As a bonus feature, a model of S-layer integration of the three types of filaments is provided.

This is an excellent manuscript. It is well written and the insights are clearly presented. The figures are excellent. A comprehensive introduction including relevant citations is provided. The results are significant for understanding of biofilm formation and motility in archaea. The interpretations of these results is sound and the discussion is refreshingly insightful and inspirational. There are no obvious flaws in the cryo-EM workflows.

This manuscript represents a conceptual advance in the field of structural biology of archaeella and should be published without delay.

Minor comments:

Line 186: "As such, highly flexible glycans can generate bulky glycan shields over hundreds of nanoseconds". This sentence should be reformulated (space and time is not clearly defined), e.g. by adding "... covering most of the filament surface within hundreds of nanoseconds as predicted by MD simulations".

Line 317: "screw-like topology of 317 *S. acidocaldarius* archaeella could in theory facilitate a tight interlocking of adjacent filaments". This is an appealing idea in the context of bundling, which may also more generally apply to other supercoiled

filaments such as bacterial flagella. However, the archaella presented in SI Fig. 11c/d are straight. Naturally occurring archaella are supercoiled as shown in SI 11b. In supercoils there is a slight difference in helical pitch between the outer and the inner side of the filament, creating a small mismatch between the tightly bundled archaella. It is not clear whether this mismatch would prevent bundling, but SI Fig. 11c/d is probably too idealized. It is more likely that bundling is simply achieved by mechanical synchronization of the supercoils and that this microscopic surface feature is not the primary cause for bundling.

Version 1:

Reviewer comments:

Reviewer #1

(Remarks to the Author)

The revised version of: "Towards a molecular picture of the archaeal cell surface" has been significantly improved. Clearly, the authors made efforts to address the reviewers' comments.

The following minor issues should be addressed prior to publication:

1. Figure 2 caption: remove from the title: "and ion binding sites"
2. In most figures, when a structure that was determined in a different study is presented, the authors indicated the PDB code. Please do this also for Supplementary Figures 8 and 11.
3. Tables 4 and 5: please report resolution up to one digit after the decimal point.

Reviewer #2

(Remarks to the Author)

The authors have satisfactorily addressed all my questions, with one minor exception:

The negative staining comparison between WT and Δ AgI3 samples presents only two archaeal flagella filaments, one per sample, hindering clear visualization of their waveform. To better assess their supercoiled form, could the authors provide additional lower magnification images displaying the full length of these filaments and include measurements of their height and pitch for both WT and Δ AgI3? From the current images, it's clear they are both supercoiled, but it's not clear how long they are and whether they have the same waveform.

Reviewers' comments (black)

Author's response (blue)

We would like to thank all reviewers for all the time and effort that they put into reviewing our manuscript. We have carefully addressed each comment and believe that the manuscript has improved significantly.

Reviewer #1 (Remarks to the Author):

In the manuscript "Towards piecing together a molecular picture of the archaeal cell envelope" Gaines et al. report the structures of several surface filaments from the crenarchaeon *Sulfolobus acidocaldarius* with an emphasis on the surface glycans attached to these filaments and how the filaments protrude the S-layer. The main new finding described in this manuscript is the structure of the archaellum and its "glycosylation domain". In addition to the archaellum structure, this manuscript contains a lot of data, some of which is not convincing, and some does not seem related to the main finding, such that the manuscript reads as a lot of pieces stitched together rather than one comprehensive story.

We thank the referee for their thorough review. We revised the paper for a more consistent narrative, particularly by reworking the introduction, restructuring the results section and generating new figures. The manuscript now integrates the data better, hopefully not giving the impression of being stitched together anymore.

Although reported for the first time for this organism, structures of archaella from several species have already been determined, and a subdomain rich with glycans was also seen in *S. islandicus* (Kreutzberger et al., 2022). The significance of the work as it is presented is not clear, and the work should be revised to support claims made in this manuscript (or omit these claims).

We would like to emphasize that our manuscript does not only present findings on the architecture of the archaellum, but also new insights into the structures of the Aap and thread, with and without glycan truncation. Importantly, our results demonstrate that glycan truncation does not substantially alter their architecture, clarifying a long-standing notion in the field that assumed that glycans may impact the assembly or structure of archaeal surface filaments. Furthermore, our study contextualizes the structures of these three filaments with that of the S-layer, offering fresh perspectives on filament/S-layer interactions. To our knowledge, our work thus marks the first comprehensive report on the structures of the S-layer and an array of three functionally distinct filaments from one organism. In our view, this significantly advances the field of microbiology.

Specific comments:

1. The authors use the MW2106 strain to obtain archaella. What is the mechanism of hyper-archaellation in the MW2106 strain? Could it affect filament architecture? There is no reference to the work in which this strain was generated, only a mention in Table 1 that it is from the Albers lab strain collection. If the generation of this strain was not reported earlier, its generation should be described

in the Methods and further characterization of this strain should be presented, e.g., evidence that it is hyper-archaellated and that other aspects of physiology were not severely compromised.

We have now described how the mutant was made in the M&M section. The hyper-archaellated and hypermotile phenotype of the deletion of the gene cluster was unexpected and is not understood at the regulatory level. We originally thought that these genes might be involved in thread assembly in *S. acidocaldarius*, but that does not seem to be their function. We have characterised the hyper-archaellated strain by negative stain electron microscopy (shown in the supplementary data) and motility experiments (that will be published in a separate paper). Other than observing more archaella and enhanced motility, we did not find any visual evidence that the cells are in any way compromised.

2. Cryo-EM data: a. When referring to resolution, please state that this is an average resolution of the map, as these are cryo-EM maps. b. Images of cryo-EM maps are cropped. Full images of maps with estimation of local resolution should be displayed, at least in the supplementary. c. Sample cryo-EM micrographs should be displayed in the supplementary. d. Box sizes, overlap between particles, initial helical rise and twist used in the reconstruction (not just optimized in the final reconstruction) should be mentioned in the methods for each reconstruction.

- a) Resolutions are now stated as “global” throughout.
- b) A new supplementary figure providing updated local resolution estimates have been provided. The images are no longer cropped.
- c) Sample cryoEM micrographs have been provided as another supplementary figure.
- d) The requested helical reconstruction details have been included in the M&M section.

3. The authors report updated structures of the Aap and thread. Improving resolution is always nice, but the insights gained here are not clear. For the Aap, no new finding is reported. For the thread, the authors write that they discovered a so-far unknown O-glycosylation site. How do they know that the density near T195 is a glycan and not some other modification? Could a phosphate or sulphate group fit in there, for example? Density in the cryo-EM map alone cannot determine the identity of the modification, and additional data (e.g., mass spectrometry) should be provided to support this claim.

The main advance of the improved resolutions is that surface glycans are now much better resolved. This is particularly the case for the Aap. While previously we could only partially model the glycans (Gaines et al, Nat. Commun. 2024), the new map allows us to model the full glycan. We have now stated this more clearly in our manuscript and provided new figure content to support this (new figure 4).

In addition to excellently defined N-glycans, the new cryoEM map of the thread now shows a substantial density near T195. We are confident that this density is a sugar molecule, due to its distinctive size and flattened shape that is typical for sugars, but inconsistent with sulphate or phosphate modifications (see figure below) Mass spectrometry is unfortunately not feasible, as threads resist tyrosination (Gaines et al, Nat. Commun. 2022). However, we have toned down our interpretation of this density and now refer to it as **putative** O-glycosylation site.

CryoEM map of the thread with mannose modelled (left) and phosphate modelled (right).

4. Comparison between wild-type and Δ agl3 filaments: a. The motivation for this comparison is not clear at all. The first time that glycosylation is mentioned in the introduction is in line 99: "To tackle the long standing question of how glycans impact on filament architecture...". More information on glycosylation in archaea is necessary, at the very least, a mention of the extent of surface glycosylation in archaea and why glycans might impact filament architecture. Mentioning the types of glycosylation found so far in *S. acidocaldarius* would also be helpful.

We have made this now clearer by putting more emphasis on the glycosylation in the introduction and why the impact on the structure of and function of filaments is an open question. We also clarified what types of glycosylations are found in *Sulfolobus*.

b. "Comparing these structures with those previously published for strains expressing full-length glycans 33,38 revealed no significant differences... (Figure 4)." Please be more quantitative. What is a significant (or not significant) difference? Can one calculate the RMSD between structures in certain regions? Figure 4 does not show this point convincingly. Superimposing models would be better.

We have now updated the figure (new figure 5), and show RMSD plots of ArlB and AapB monomers next to the map comparisons (Fig. 5 c-f). This should now make it clearer that the protein portion of Aap and archaella is unaffected.

c. The authors mislead the reader by writing that they solved the structure of the thread from the mutant (abstract, results, discussion). In the Methods they write that they built a model of the thread from the mutant. However, a model was not built (=structure was not solved), as revealed by the authors after I requested the model, map, and validation report for this structure. Moreover, the authors report a 4.1 Å resolution for this map, but when receiving the map and looking at it, it is clear that this is an overestimation. The quality of the map is poor. It is blobby, with no secondary structure observed, while secondary structure and some side-chain density should be visible at the reported resolution. With this kind of data, I do not understand how the authors conclude that the structure of the thread is unaffected by the lack of the Sulfoquinovose (line 221). The authors should either

improve the map, build a model, and make a proper comparison, or omit this entire part from the manuscript.

We appreciate the referee's analysis. Reinvestigating the Δ agl3 thread data, we agree that the global resolution indicated by the FSC was indeed an overestimation. We attempted to improve the map from the data that we have but were unable to do so. This is likely due to excessive bundling of the Δ agl3 threads, which prohibited proper alignment. As we would like to avoid the time-consuming and expensive collection of new data for a likely negative result, we have decided to remove the Δ agl3 thread data from our manuscript and instead focus on analysing the motile filaments, namely the aap and archaella, for which high resolution was obtained. This makes sense, as the aim was to investigate if the previously reported glycan ablation on motility is linked to defects in the motility-conferring structures.

5. Metal-binding site: a. The text refers to this issue after presenting figs. 3 and 4, yet the data is presented in fig. 2c. This is very confusing. I recommend moving this figure to the supplementary. b. Fig. 2c is tiny and it is extremely hard to see the metal ion depiction. c. "Revisiting the archaella structures from *P. furiosus* and *M. hungatei*, we find that the *P. furiosus* archaellum also contains the same metal coordination site". Revisiting means the authors inspected again the cryo-EM map? If so, the density of the metal ion should be shown. d. Hydrogen bond numbers are reported for different structures. How was this calculation done? **Is the calculation normalized to the number of residues?** In *S. acidocaldarius* there are more residues, thus the number of hydrogen bonds is expected to be higher. So, is the argument here that larger archaellins are more stable than smaller ones?

a) We fixed this issue by restructuring our narrative

b) We have removed the figure panel on the metal binding site from Figure 2. As we think that this is an important point, we dedicated a new figure (Figure 3) to the metal binding site. The panels are enlarged, so it should now be easier to see the details.

c) We agree that the wording was confusing. We revisited the *P. furiosus* atomic model and found that the metal coordination site exists there as well. However, the map was of insufficient resolution (4.1 Å) to reveal these metal ions. We now reworded the relevant sentence to make this clearer.

d) We appreciate that this point was rather vague. We investigated the archaellins with and without metal binding sites more closely and found that the lack of a stabilising ion in loop Gly240 – Ser253 in sulfolobales is likely compensated by an additional β -hairpin (Phe 150 - Tre 161). This β -hairpin is part of the glycosylation sub-domain and packs against the ion-less loop. Additional stability is provided by hydrogen bonds formed between the β -hairpin 150-161 and loop 240-253. We present these findings via the new figure 3 and a new paragraph in the results section. H-bonds were visualised in ChimeraX.

6. Some figures are not clear and hardly convey the message written in the text/caption. Examples: 1. Supplementary Figs. 3 and 9, and especially Fig. 4d, are hard to interpret. Why is the mesh in Fig. 4d shown differently for wt and the mutant (more dense and smooth for Δ agl3)? I recommend making the mesh partially transparent, so that it is easier to see the model, and to remove from the figure regions of the map and model that are irrelevant to what the authors are trying to show. 2. " The S.

acidocaldarius thread contains a glycan linked to N146, which is wedged in between neighbouring subunits, thus likely restricting both the flexibility of the glycan as well as that of the filament 33 (Figure 9 c)." Should be supplementary Figure 9c. Also, this point is not clear at all from this figure. It would be helpful to color two neighboring subunits differently and show that the glycan is wedged between them.

All these figures have been redesigned to visualise the glycans more clearly. We also corrected the labelling of the glycans, as well as an error regarding the thread glycans. In fact, it is glycan Asn56 and Asn83 that sit in between adjacent subunits. Asn146 is wedged into a cleft of the same subunit. While Asn146 is most restricted in its movement, it is likely that all three glycans restrict filament flexibility.

7. Since Nature Communications is not a specialized journal for archaea, please explain the term "crenarchaeota".

Crenarchaeota and euryarchaeota are now briefly introduced in the introduction.

8. The first paragraph of the introduction contains a lot of details on archaella components, which is hard to follow. Is all this information necessary? If so, an illustration in the supplementary would help the reader to follow.

We have made a new supplementary fig. 1, to aid this part of the introduction.

9. In the introduction, the authors explain that several archaellins may be encoded, depending on the species. However, there is no mention of how many, and which are encoded in *Sulfolobus acidocaldarius*. Later, they conclude that ArlB is the only component of the archaellum based on the cryo-EM map. Are there any other archaellins that did not fit the map?

We have now mentioned in the introduction that only one archaellin is encoded in the archaellum operon of *S. acidocaldarius*. However, some pilin subunits of T4P are encoded far away from the rest of the machinery. Thus, only the structure is the final proof for which and how many subunits compose a filament. An example is Neuhaus et al, 2020, who found an unexpected pilin in *Thermus thermophilis* through structural analysis. To avoid confusion, we have reworded the sentence in question as follows: *Based on our cryoEM map, we built an atomic model ab initio. The global resolution of 2.0 Å enabled us to easily recognise and model most amino acids manually, confirming that the archaellum is composed of the archaellin ArlB.*

10. Fig. 2b: Doesn't look like all images are on the same scale (For example, the purple one looks bigger than others).

The figure panel was remade from one ChimeraX session displaying all structures side-by-side. The structures are thus all at the same scale now.

11. The captions of supplementary Figs. 7,8 report a scale bar, but I cannot find a scale bar in these figures.

The scale bars have now been added.

12. "Interestingly, only the O4-linked α -mannose is missing from the glycan tree, while the O6-linked α -mannose is still present... This may indicate that the two mannose molecules are added by different enzymes." Why is this interesting (as stated in the beginning) or new? Wasn't this already resolved in Meyer et al., 2011? The structure is just consistent with the earlier finding.

The enzymes catalysing the addition of the two mannose molecules to the N-glycan in *S. acidocaldarius* are currently unknown.

13. There are two instances where the authors write that they investigated AlphaFold2 prediction models, yet no figure/table/text shows this data (metal binding site and glycosylation domain).

For both instances, Alphafold models have now been included in the main and supplementary figures and have been clearly referenced in the results.

14. Supplementary Fig. 11: a. The caption states: "xxxx strain". b. It is not clear from the caption what the difference is between a and b, other than the magnification. c and d. Based on what were these maps/models placed next to each other at these positions? It is impossible to see from these views that ridges and notches interlock and that residues do not clash with each other.

We have removed the hypothesis regarding interlocking archaella from the manuscript, as we have come to the conclusion that this is too speculative at this point.

15. Fig. 7: the caption states that b shows the archaellum and c the Aap, but the figure shows the opposite.

Corrected.

16. In the discussion: "While archaella have been seen to bundle in multiarchaellated archaea 66–68, *S. acidocaldarius* has few archaella per cell and archaella bundling is not observed in swimming motion." The authors cite a paper in which archaella bundling is observed in swimming motion of *Halobacterium salinarum*. Is there a paper showing directly, with a comparable technique, that the archaella of *S. acidocaldarius* do not bundle in swimming motion? If so, it should be cited. If not, the authors should refrain from making such a statement.

As mentioned above, we have removed the hypothesis regarding interlocking archaella from the manuscript.

17. The presentation could be improved. There are several typos, changes in font in the supplementary, inconsistency in naming the mutant strain (Δ agl, Δ agl, Δ Agl), etc.

Font sizes and labels are now consistent.

Reviewer #2 (Remarks to the Author):

In this article, Gaines et al provided an in-depth structural analysis of the protein filaments from *Sulfolobus acidocaldarius*, an archaeal species known for its complex motility and adhesion mechanisms. Using cryo-electron microscopy, the study achieves high-resolution structures of the archaellum and other associated filaments, including novel insights into their N-glycan modifications and architectural dynamics. These findings deepen our understanding of archaeal cell envelope structures and their functional implications, highlighting distinct glycosylation patterns that influence filament stability and motility. Overall, the story is good, and my co-reviewer and I have some issues that need to be addressed in a revision.

Major issues:

(1) The authors used the hyper-archellated strain MW2106 to express high amounts of archaella for cryoEM. Are they able to collect electron cryo-tomography data of one of these cells? This could be really cool if they are able to visualize emerging filaments and relate their position to the S-layer lattice.

We thank the reviewer for their suggestion. This is indeed something we are currently working on. However, we would prefer to include these data in a future paper with a new focus.

(2) The N-glycosylation observation is great. Could the authors provide figures that adjust the map threshold to clearly show the extra glycan densities that come from Asn but not adjacent S/T? This would be very helpful for the readers to immediately understand how you know it's N-linked glycosylation.

The figures showing glycosylation have been redone, and in each case, a bond between the Asn can be seen to clearly connect to the sugar, thus removing any doubt of an O-glycosylation through Ser/Thr linkage. We have also revamped figure 4, which now focusses on glycosylation and shows maps and models more clearly.

(3) The authors mention that "S. acidocaldarius glycans are tri-branched hexasaccharides." Given that N-linked glycosylation is generally better understood than O-linked glycosylation, it would be beneficial for the authors to use mass spectrometry to validate the components of these glycans. If this is not feasible, a rationale for its impracticality should be provided.

Additionally, in Fig. 4d, where glycans are included in the model, the authors need to clarify whether these are merely illustrative or if they were actually incorporated into the structural model. If it's the latter, supporting mass spectrometry data should be included to validate the glycan structures depicted.

The composition of the glycans has previously been determined using mass spectrometry (Peyfoon et al, 2010; Meyer et al, 2011) and are consistent with the glycan densities that we

have observed in s-layers and filaments of *S. acidocaldarius* (Gaines et al, 2022; Gambelli et al, 2022 and Gambelli et al, 2024). We have now stated this more clearly in the manuscript.

(4) The impact of the Δ Agl3 mutation on motility is intriguing. However, it remains unclear whether this mutation significantly affects the morphology of the flagella, such as supercoiling or the number of filaments produced. To clarify this, the authors should include negative staining images that compare the morphology of the flagella from both wild-type and mutant strains. This would help determine if the observed changes in motility correlate with structural alterations in the flagella.

We have recorded negative stain images of the mutant and wild type and see no obvious differences in the morphology of the cells or the supercoiling or number of filaments produced. We have included example images as a new supplementary figure 15.

(5) I noticed that the term "archaellum" is also referred to as "archaeal flagellum" in some other papers. While I personally have no preference for the terminology used, it can be confusing for readers unfamiliar with this field. After randomly asking several colleagues who do not specialize in archaea or flagella-related topics, it appears that "archaeal flagella" is more easily understood. Although it is too late to change the terminology in the field, could the authors clarify the first mention of "archaellum" to indicate that it is also known as "archaeal flagellum"? This small addition could help improve clarity for a broader audience.

We have clarified the term archaella on its first use (see abstract).

Minor issues:

Line 330, is there a reference supporting the described relationship between filament dimensions and mechanical forces?

We have reviewed the literature for how filament width influences hydrodynamics and swimming efficiency more deeply and consulted a physicist specialising in hydrodynamics at the microscopic scale. It appears that the gained thrust would be marginal, considering that the increased surface area of the wider filament would also increase drag. It is more likely that widening the filament's diameter would stiffen it, thus influencing the waveform of the supercoil. We have changed the narrative accordingly and included appropriate references.

Line 336-341, Author's have not mentioned here that Aap in *S. islandicus* has been shown to exist in two conformations, one which has a tri-conformer arrangement and the other with a mono-conformer. These two conformers have opposite orientations of the C-terminal globular domain with respect to the N-terminal helical domain. In light of the author's suggestion that the 3-conformer could support rapid retraction, might they mention these two distinct forms? This evidence from *S. islandicus* seems relevant to discussion of rapid retraction if, for instance, the two states are conformational exchangeable (one representing pre-retraction, the mono-conformer, and the other representing active retraction or post-pre-retraction).

We thank the reviewer for their suggestion and incorporated it into our results and discussion sections.

Structure Table. Why is sharpening B factors is 0? What is Model Refinement resolution? I've never heard this term before. Model resolution: model is atom coordinates, and it doesn't have a resolution. You should report this as model: map FSC (0.5) and put it under "map resolution estimation".

Thank you for pointing this out to us. We found that maps without sharpening (sharpening B-factor 0) and without post-processing were the most suitable for modelling glycans, which are less ordered compared to the protein part of the filament. Initially, we attempted to use maps sharpened in cryoSPARC, via DeepEMhancer, and EMRready for glycan modelling; however, these methods resulted in the loss of critical features.

We agree that "model" resolution" is confusing and have renamed it map/model resolution. The row "Model refinement resolution" was inherited from earlier papers and included based on the map resolution from data processing used in Refmac5/Phenix. However, this is now redundant, as we also quote map/model resolution. We have therefore deleted it now.

I didn't see any raw micrographs (cryo-EM, not negative staining) or 2D averages showing what those three different filaments look like. This is helpful information to include somewhere in the paper.

A new supplementary figure displaying the raw micrographs has now been added.

Reviewer #3 (Remarks to the Author):

Reviewer #4 (Remarks to the Author):

Gaines et al present 3 types of archaeal filaments from *S.acidocaldarius* at resolutions up to 2 Angstroms – the highest achieved to date for these archaella, Aap and threads. This enabled the authors to identify and model glycans in all three filaments, identifying a novel β -linked glycosylation site. As a bonus feature, a model of S-layer integration of the three types of filaments is provided.

This is an excellent manuscript. It is well written and the insights are clearly presented. The figures are excellent. A comprehensive introduction including relevant citations is provided. The results are significant for understanding of biofilm formation and motility in archaea. The interpretations of these results is sound and the discussion is refreshingly insightful and inspirational. There are no obvious

flaws in the cryo-EM workflows.

This manuscript represents a conceptual advance in the field of structural biology of archaea and should be published without delay.

We thank the reviewer for their very positive evaluation of our manuscript.

Minor comments:

Line 186: “As such, highly flexible glycans can generate bulky glycan shields over hundreds of nanoseconds”. This sentence should be reformulated (space and time is not clearly defined), e.g. by adding “... covering most of the filament surface within hundreds of nanoseconds as predicted by MD simulations”.

Many thanks for the suggestion - we have reworded this sentence accordingly.

Line 317: “screw-like topology of 317 *S. acidocaldarius* archaea could in theory facilitate a tight interlocking of adjacent filaments”. This is an appealing idea in the context of bundling, which may also more generally apply to other supercoiled filaments such as bacterial flagella. However, the archaea presented in SI Fig. 11c/d are straight. Naturally occurring archaea are supercoiled as shown in SI 11b. In supercoils there is a slight difference in helical pitch between the outer and the inner side of the filament, creating a small mismatch between the tightly bundled archaea. It is not clear whether this mismatch would prevent bundling, but SI Fig. 11c/d is probably too idealized. It is more likely that bundling is simply achieved by mechanical synchronization of the supercoils and that this microscopic surface feature is not the primary cause for bundling.

Agreed. We have concluded that this point is perhaps too speculative (and indeed idealised) and have removed this discussion point from our manuscript.

Reviewers' comments (black)

Authors' response (blue)

Reviewer #1

1. Figure 2 caption: remove from the title: "and ion binding sites"

Done. Please note that I have given Figure 2 a new title to reflect the content more comprehensively:

Figure 2: Structures of archaella in comparison

2. In most figures, when a structure that was determined in a different study is presented, the authors indicated the PDB code. Please do this also for Supplementary Figures 8 and 11.

Done

3. Tables 4 and 5: please report resolution up to one digit after the decimal point.

Done

Reviewer #2

The authors have satisfactorily addressed all my questions, with one minor exception:

The negative staining comparison between WT and Δ Agl3 samples presents only two archaeal flagella filaments, one per sample, hindering clear visualization of their waveform. To better assess their supercoiled form, could the authors provide additional lower magnification images displaying the full length of these filaments and include measurements of their height and pitch for both WT and Δ Agl3? From the current images, it's clear they are both supercoiled, but it's not clear how long they are and whether they have the same waveform.

We have provided a new supplementary figure (16) that shows example images of whole archaella emerging from WT and Δ Agl3 cells. In the micrographs, the apparent length and waveform of the archaella varies significantly, even within the same sample. Therefore, we cannot draw any clear conclusions about changes in waveform and length between WT and the mutant.